

# Arc volcanism, carbonate platform evolution and palaeo-atmospheric CO₂: Components and interactions in the deep carbon cycle

Jodie Pall[1], Sabin Zahirovic[1], Sebastiano Doss[1], Rakib Hassan[1,2], Kara J. Matthews[1,3], John Cannon[1], Michael Gurnis[4], Louis Moresi[5], Adrian Lenardic[6] and R. Dietmar Müller[1]

[1]EarthByte Group, School of Geosciences, University of Sydney, NSW 2006, Australia.
[2]Geoscience Australia, GPO Box 378, Canberra 2601, ACT, Australia
[3]Department of Earth Sciences, University of Oxford, Oxford, OX1 3AN, UK.
[4]Seismological Laboratory, California Institute of Technology, California 91125, USA.
[5]School of Earth Sciences, University of Melbourne, Victoria 3010, Australia.
[6]Department of Earth Science, Rice University, Texas 77005, USA.

*Correspondence to*: Jodie Pall (jodierae.pall@gmail.com)

## Abstract

Carbon dioxide ($CO_2$) liberated at arc volcanoes that intersect buried carbonate platforms plays a larger role in influencing atmospheric $CO_2$ than those active margins lacking buried carbonate platforms. This study investigates the contribution of carbonate-intersecting arc activity on palaeo-atmospheric $CO_2$ levels over the past 410 million years by integrating a plate motion model with an evolving carbonate platform development model. Our modelled subduction zone lengths and carbonate-intersecting arc lengths approximate arc activity with time, and can be used as input into fully-coupled models of $CO_2$ flux between deep and shallow reservoirs.

Continuous and cross-wavelet as well as wavelet coherence analyses were used to evaluate trends between carbonate-intersecting arc activity, non-carbonate-intersecting arc activity and total global subduction zone lengths and the proxy-$CO_2$ record between 410 Ma and the present. Wavelet analysis revealed significant linked periodic behaviour between 75-50 Ma, where global carbonate-intersecting arc activity is relatively high and where peaks in palaeo-atmospheric $CO_2$ is correlated with peaks in global carbonate-intersecting arc activity, characterised by a ~32 Myr periodicity and a 10 Myr lag of $CO_2$ peaks after carbonate-intersecting arc length peaks. The linked behaviour may suggest that the relative abundance of carbonate-intersecting arcs played a role in affecting global climate during the Late Cretaceous to Early



Paleogene greenhouse. At all other times, atmospheric $CO_2$ emissions from carbonate-intersecting arcs were not correlated with the proxy-$CO_2$ record. Our analysis did not support the idea that carbonate-intersecting arc activity is more important than non-carbonate intersecting arc activity in driving changes in palaeo-atmospheric $CO_2$ levels. This suggests that tectonic controls are more elaborate than the

subduction-related volcanic emissions component or that other feedback mechanisms between the geosphere, atmosphere and biosphere played larger roles in modulating climate in the Phanerozoic.

## 1 Introduction

The current paradigm of the deep carbon cycle (I.e. the planetary cycling of carbon over million-year timescales) attributes fluctuations in the atmospheric carbon dioxide ($CO_2$) to the realm of tectonic forces,

where arc-volcanic emissions (Van Der Meer et al., 2014; Kerrick, 2001) and metamorphic decarbonation (Lee et al., 2013) are major $CO_2$ sources, and the processes of silicate weathering (Sundquist, 1991; Kent and Muttoni, 2008) and marine organic carbon burial (Berner and Caldeira, 1997; Ridgwell and Zeebe, 2005) are major sinks removing $CO_2$ from the atmosphere. Subduction plays a critical role in this cycle. In a dynamic interplay, oceanic lithosphere is consumed at subduction zones, removing carbon bound in

pelagic carbonate seafloor sediments from the exogenic carbon cycle. Simultaneously, $CO_2$ is emitted during arc volcanism and at mid-ocean ridges (MORs) where new oceanic crust is being created (Burton et al., 2013). Despite the proposal that silicate weathering has, at some stages, been a dominating control of atmospheric $CO_2$ levels (Kump, 2000; Kent and Muttoni, 2013), arc magmatism at icehouse-greenhouse transitions is thought to be the first-order control on climate fluctuations while silicate

weathering acts to modulate atmospheric $CO_2$ as a secondary regulative process (Ridgwell and Zeebe, 2005; Lee and Lackey, 2015; McKenzie et al., 2016). Recent studies have found support for links between global arc activity and icehouse-greenhouse transitions using detrital zircon ages, modelling and experimental techniques, particularly as drivers of greenhouse conditions in the Cambrian (McKenzie et al., 2016; Cao et al., 2017), Jurassic-Cretaceous (McKenzie et al., 2016) and early Paleogene (Lee et al.,

2013; Carter and Dasgupta, 2015; Cao et al., 2017).



Recently, carbon and helium isotope analysis from modern volcanic arc gas has provided evidence that volcanic arcs that assimilate crustal carbonate in their magmas through decarbonation reactions have a greater atmospheric $CO_2$ contribution than other types of arcs (Mason et al., 2017). Modern crustal carbonate reservoirs are a result of global organic and inorganic carbonate production throughout the

Phanerozoic, which has contributed to the build-up of expansive, shallow-water carbonate sequences including ramps and rimmed shelves along continental margins, hereafter referred to as 'carbonate platforms' (Kiessling et al., 2003). At different points in Earth's history, emissions from continental, carbonate-intersecting arcs (CIAs) may have dominated global volcanic carbon flux, such as during the Cretaceous (144-65 Ma) where continental arcs are hypothesised to have been longer than modern day

lengths and where a greenhouse climate prevailed (Lee et al., 2013; van der Meer et al., 2014; Cao et al., 2017). It is plausible that crustally-derived carbon at other points in Earth's history has played a significant role in affecting global climate (Lee and Lackey, 2015), yet it is difficult to test this hypothesis, especially due to preservation bias in the geological record (Berner, 2004).

We aim to investigate whether volcanic arcs that interact with buried carbonate platforms (CIA activity)

are significantly different to volcanic arcs that do not intersect with buried carbonate reservoirs (non-carbonate-intersecting arc [NCIA] activity) in influencing global palaeo-atmospheric $CO_2$ concentrations. Our study explores volcanic arc activity as an aspect of the deep Earth carbon cycle using a combination of plate reconstruction software and wavelet analysis. Only Cao et al. (2017) and this study have endeavoured to look beyond the use of palaeogeographic maps to explore the distribution of continental

arcs through space and time in a plate tectonic framework. We model space-time trends of global subduction zone and carbonate platform interactions since the Devonian (410 Ma), where subduction zone lengths are used as a surrogate measure for volcanic arc lengths. This paper is accompanied by an open-source 'subduction zone analysis toolkit' (Doss et al., 2016) to help enable the use of digital plate motion models (e.g. Matthews et al., 2016) and other tools to investigate aspects of the deep carbon cycle.



## 2 Modelling global arc activity and carbonate platform evolution

We investigate the combined effect of global volcanic arc activity and the subsequent decarbonation of crustal carbonates by arc magmas to create a first-order approximation of the contribution of carbonate-intersecting arc volcanism to palaeo-$CO_2$ levels over the past 410 Myr. An

'Accumulation Model' of carbonate platform evolution was produced to represent the long-term persistence and build-up of carbonate platforms in upper continental crust. Global subduction zone lengths are used as a surrogate for global volcanic arc lengths through time, where we assume a one-to-one correspondence of arc lengths to volcanic activity. Using open-source and cross-platform plate reconstruction software GPlates (http://gplates.org/download.html) and a new pyGPlates Python

application programming interface (API) (https://www.gplates.org/docs/pygplates/), we track the interaction between subduction zones and major carbonate platforms through time to better understand the effect of carbonate-intersecting arc magmatism on palaeo-$CO_2$ and palaeoclimates interpreted from proxy data (Park and Royer, 2011).

### 2.1 An 'Accumulation' model of carbonate platform development

Palaeogeographic maps of carbonate platforms spanning the Phanerozoic from Kiessling et al. (2003) were used to assemble a time-dependent evolution of carbonate platform accumulation. The twenty-four maps represent the spatial extent of carbonate platforms at the maximum marine transgression within each epoch for the Phanerozoic (Kiessling et al., 2003). The geometries were georeferenced to geographic WGS84 co-ordinates in ArcGIS (ESRI, 2011) to be compatible with GPlates, the open-source plate

reconstruction software (Boyden et al., 2011).

A GPlates file was created to depict continuously evolving carbonate platform development from 410 Ma to present-day. Creating this model involved stitching together twenty-four carbonate platform shapefiles, where reef patterns in each epoch had been adjusted to the Golonka and Kiessling (2002) Phanerozoic timescale. However, owing to new chronostratigraphic data, the geological times ascribed to the existence

of carbonate platforms in each epoch were converted to the corresponding times in the latest chronostratigraphic scheme given in the 2016 version of the International Stratigraphic Chart (Cohen et





al., 2013). Subsequently, the static carbonate platform shapefiles were reconstructed with the rotations and plate geometries associated with an older plate model (Scotese, 2016), similar to the model used by Kiessling et al. (2003). In GPlates, the carbonate platform geometries were assigned Plate IDs based upon their overlapping position within the reconstructed continental geometry.

5    Carbonate platforms were rotated to their present-day positions, and Plate IDs associated with the Scotese (2016) model were translated into corresponding Plate IDs from the Matthews et al. (2016) plate motion model. By obtaining the present-day geometry of the ancient carbonate platforms, attaching these geometries to any other plate motion model becomes straightforward. The evolving carbonate platform model was created by layering each carbonate platform geometry such that it persists to present-day. The

10   'Accumulation' Model we implement in this study embodies the idea that carbonate platforms accumulate in crustal reservoirs through time.



**Figure 1:** (a) Schematic representation of the *grdtrack* tool. Using *grdtrack*, ~450-km-long cross-profile 'whiskers' were constructed along subduction zones (red and black) spaced 10 km apart in the direction of subduction. At regular intervals along the whiskers, the carbonate platform grid on the overriding plate was sampled where carbonate polygons have a value of 1, and everywhere else is zero. If the whisker intersects with a carbonate platform, it represents 10 km along a carbonate-intersecting volcanic arc. (b) Schematic representations of carbonate-intersecting continental arcs. Carbonate platforms become buried over time, forming reservoirs in the crust. Through assimilation and decarbonation reactions, arc magmas interact with upper-crustal carbonates and liberate significantly more $CO_2$ emissions that at non-intersecting continental arcs.

## 2.2 Assumptions and limits of the Accumulation Model

The Accumulation Model is an end-member scenario of how carbonate platforms evolve through time. Carbonate accumulation is assumed to accrete onto continental margins and persist as mid-crustal carbonate reservoirs from deposition to the present (Fig. 1). Ancient carbonate platforms are known to



persist to present-day in surface reservoirs, and can be reconstructed from the geological record as they either outcrop, are sampled by drilling or are interpreted from seismic reflection studies (Kiessling et al., 2003). Some crustal carbonate is inevitably eroded or subducted into the deep mantle, however it is improbable that most reservoirs have been destroyed in the way as most carbonate platforms accumulate

on continental margins and are not likely to subduct (Lee and Lackey, 2015). Given limited erosion on passive continental margins, except in the cases of major uplift and deformation through mountain-building, it is far more likely that carbonate platform expansion has exceeded their depletion by erosion through time (Ridgewell and Zeebe, 2005). Moreover, the existence of fossil reef data as far back as the Ordovician (Kiessling et al., 2003) favours the notion that extensive portions of platforms have been well-

preserved in continents.

Since our model is temporally limited to the Devonian, it suggests that carbonate-arc intersections would have increased through time, which is likely an oversimplification of the carbonate depositional process on continental margins. In addition, since the accumulation rate of carbonate platforms is variable in both space and time, we apply a simplistic Boolean assumption (i.e., a carbonate platform is likely to persist

in the sedimentary record, with an unknown thickness).

A significant limitation of the Accumulation Model is that we assume carbonate platforms have not been significantly depleted by sustained volcanic interaction through time. We follow this assumption because there is no way to account for their rate of depletion given the complexity of inter-dependent factors such as heat, pressure, composition and the duration of interaction (Johnston et al., 2011). The model also does

not account for the subduction of some carbonate platforms over the past 410 Ma, although it is expected that accounting for this would not drastically change the results. Finally, because subduction of carbonate platforms is not considered, the Accumulation Model provides an upper bound of carbonate-arc interactions. However, this may be a reasonable proxy for the role of subducted carbonate platforms that would contribute substantial volatiles into the mantle wedge above a subducting slab.



## 2.3 Measuring global subduction zone lengths with pyGPlates

We use the open access global plate motion model from Matthews et al. (2016) to analyse the spatio-temporal distribution of subduction zones in a deep-time tectonic framework and test the hypothesis that subduction zone lengths are correlated with atmospheric $CO_2$ levels. The model spans much of the

Phanerozoic as it links the 410-250 Ma Domeier and Torsvik (2014) and the 230-0 Ma Müller et al. (2016) plate motion models. The pyGPlates library enables access to common GPlates functions using the Python programming language, a framework that facilitates model analysis and data processing. Using the pyGPlates workflow (Doss et al., 2016), global subduction zone boundaries were extracted from the Matthews et al. (2016) plate motion model in 1-million-year time steps from 410 Ma to present-day.

Importantly, subduction zone geometries were adjusted to remove any overlapping line segments that would overestimate arc lengths through time.

## 2.4 Computing the intersections of carbonate platforms and subduction zones

We test whether volcanic arcs interacting with carbonate platforms are different from non-intersecting arcs in their contribution to global atmospheric $CO_2$ concentrations, and hence we track the lengths of

subduction zones that both do and do not intersect with carbonate platforms in the overriding plate. Carbonate platform geometries were converted to polygons using *grdmask* from the development version of Generic Mapping Tools (GMT) (v.5.2.1; Wessel et al., 2015) which created a Boolean-style mask consisting of closed domains with a value of 1 where carbonate platforms existed and 0 elsewhere. We identified polygons proximal to subduction boundaries through time using the exported subduction zone

geometries from the pyGPlates workflow. We compute the signed-distance function, positive toward the overriding plate, on the surface of the sphere for all subduction zones and computed where the carbonate platforms lie within +448 km of the trench. The *grdtrack* tool in GMT was used to create large tracks of ~450 km-long cross-profiles perpendicular to subduction boundaries at a uniform spacing of 10 km, with 'whiskers' pointing in the direction of subduction (Fig. 1a). The whiskers had five evenly-spaced nodes

to detect intersections with carbonate platform polygons. We determined the ~450 km-long inclusion distance by determining average arc-trench distances using the present-day 'Volcanoes of the World'




database maintained by the Smithsonian Institution's Global Volcanism Program (Siebert and Simkin, 2014). From a sample size of 1023 volcanoes, average arc-trench distances are 287 km with a standard deviation of 161 km, which gives an upper estimate distance of 448 km. This captures ~84% of the location of all volcanic arcs and corresponds to an upper limit by which carbonate platforms can interact

with volcanic arcs in our assumptions. The whiskers function as a buffer, allowing us to identify the lengths of subduction zone boundaries within ~450 km perpendicular radius of a carbonate platform polygon. Cross-profiles that overlap with carbonate platform polygons demarcate areas where arc volcanoes interact with crustal carbonates (Fig. 1b). Due to the complexity and time-variability of subduction, we do not consider times during which flat slab subduction may have occurred, which would

result in greater arc-trench distances and is beyond the scope of this study.

## 3 Linking arc volcanism to palaeoclimate change

Relationships between oscillations in two time-series can be examined using a wavelet analysis to elucidate the scales and time intervals where the proxy record and modelled data display correlated, periodic signals. We performed wavelet analysis as a means of identifying localised variations of

power between proxy records of $CO_2$ and modelled carbonate-intersecting arc activity and in turn investigate the temporal evolution of these aperiodic signals. We are able to carry out this analysis as the geophysical data exhibit non-stationarity; dominant periodic signals vary in their frequency and amplitude through time. The continuous wavelet transform (CWT) is particularly useful for this task as it better characterises oscillatory behaviours of signals than discrete wavelet transforms.

The CWT was applied to decompose arc lengths and proxy signals into time and frequency space simultaneously using functions (wavelets) that vary in width to discern both the high and low frequency features present (Lau and Weng, 1995; Supplement 3). To examine whether trends in atmospheric $CO_2$ and carbonate-intersecting arcs are connected in some way, the cross-wavelet transform (XWT) was carried out between the proxy-$CO_2$ data and each of the modelled data on

carbonate-intersecting arc lengths (Supplement 4). The XWT reveals space-time regions where the two datasets coincidentally have high common power as well as calculating the phase



relationships between signals, indicating where two series co-vary (Grinsted et al., 2004). Wavelet coherence (WTC) is computed in tandem with the XWT and determines if the time series are significantly interrelated in the frequency domain.

All wavelet analysis was carried out for each detrended and filtered time series using the Wavelet
Coherence Toolbox™ for MATLAB® based on the statistical approach applied to geological and geophysical data by Torrence and Compo (1998) and Grinsted et al. (2004).

### 3.1 Data filtering

The modelled results and proxy-$CO_2$ time series were pre-processed for wavelet analysis by detrending and applying a low-pass filter using MATLAB®'s Signal Processing Toolbox™. The proxy-$CO_2$ time
series is non-uniformly distributed with multiple data values at certain time steps. Prokoph and Bilali (2008) stress the importance of keeping a consistent time-scale across all time-series when examining causation between trends. Therefore, the data were first processed so that the median atmospheric $CO_2$ (ppm) value was taken at all time steps with multiple observations. The data were then resampled with the MATLAB *resample* function to have 410 uniformly spaced data points across the 410 Myr
timeframe (Fig. S1). The function interpolates the time series linearly, and the series was found to be relatively insensitive to the interpolation method used. A moving average filter was applied using the MATLAB *filter* function with a window size of 7 to remove high-frequency noise (Fig. S2). In the time-frequency domain, our focus is on the power in the mid-to-high frequency spectrum and so the moving average filter was designed to remove periods less than 5 Myrs. As a result, the analysis does not
capture any geologically-rapid (few Myrs) changes in $CO_2$ that may be driven by changes in plate tectonics and carbonate platform-arc interactions.

Rather than comparing modelled data with Phanerozoic atmospheric $CO_2$ models like COPSE (Bergman et al., 2004) or GEOCARBSULF (Berner, 2006), results were compared to the proxy-$CO_2$ record compiled by Park and Royer (2011) because the unresolved uncertainties in model
calculations may have rendered comparisons less meaningful than when comparing with proxy records. The proxy-$CO_2$ dataset presented by Park and Royer (2011) is an expanded version of the





Royer (2006) collection, which originally incorporated data from 490 sources from the Phanerozoic (542 Ma to present) to present including palaeosols, foraminifera, stomatal indices, $\delta^{11}B$ in marine carbonates and the $\delta^{13}C$ of liverworts. The collection was chosen as it is the most up-to-date proxy-$CO_2$ data with the highest temporal resolution available.

## 4 Results

### 4.1 Continuous Wavelet Transform (CWT)

CIA lengths exhibit ephemeral mid- to long-wavelength periodicities that appear to dominate for prolonged time intervals, yet are completely absent during others. A significant periodic component in the 24-48 Myr band exists from 300 to 350 Ma, however, the signal disappears in the following time period between 300 and 225 Ma (Fig 2a). The 24-48 Myr periodic signal returns between 225 to 125 Ma, transitioning into the 48-60 Myr band between 150 and 100 Ma. Between 100 and 50 Ma, significant power in the signal returns in the 14-48 Myr band. Wavelet power is highest in wavebands > 80 Myrs, as an 80-100 Myr cyclic signal becomes significant between 190 and 110 Ma, remembering that the spectrum outside the Cone of Influence (COI) must be excluded from analysis due to distortion. In contrast, there are broad regions where signals of mid- to long-wavelength components remain strong in the NCIA length data (Fig. 2b). Notably, the strongest significant signal components are in the 48-96 Myr band in the 325-125 Ma time period and the 24-32 Myr band in the 325-100 Ma time period, demarcating a broad region of power where, conversely, there is distinctly no signal component in the CIA length signal (Fig. 2a). At certain time periods the 10-18 Myr signal component in NCIA lengths becomes significant, which is not identifiable in CIA data.

Intermittent, sporadic regions of short- to mid-wavelength signal components appear in the proxy-$CO_2$ record, with the highest peaks within a confidence level of 95% occurring between 350 to 250 Ma (Fig. 2c). During this time period, a prominent peak of ~32 Myr cycles appears, transitioning into a broader 12-36 Myr band between 300 and 250 Ma (Fig. 2c). No other waveband in the proxy-$CO_2$ record is characterised by significant spectral power within the COI. Total global subduction zone lengths have a number of clear and persistent signals through time, primarily in the 24-32 Myr, 48-64 Myr and 100-



128 Myr waveband (Fig. 2d). The long-term, 100-128 Myr periodicity is not well localised in time as much of the result lies outside the COI.





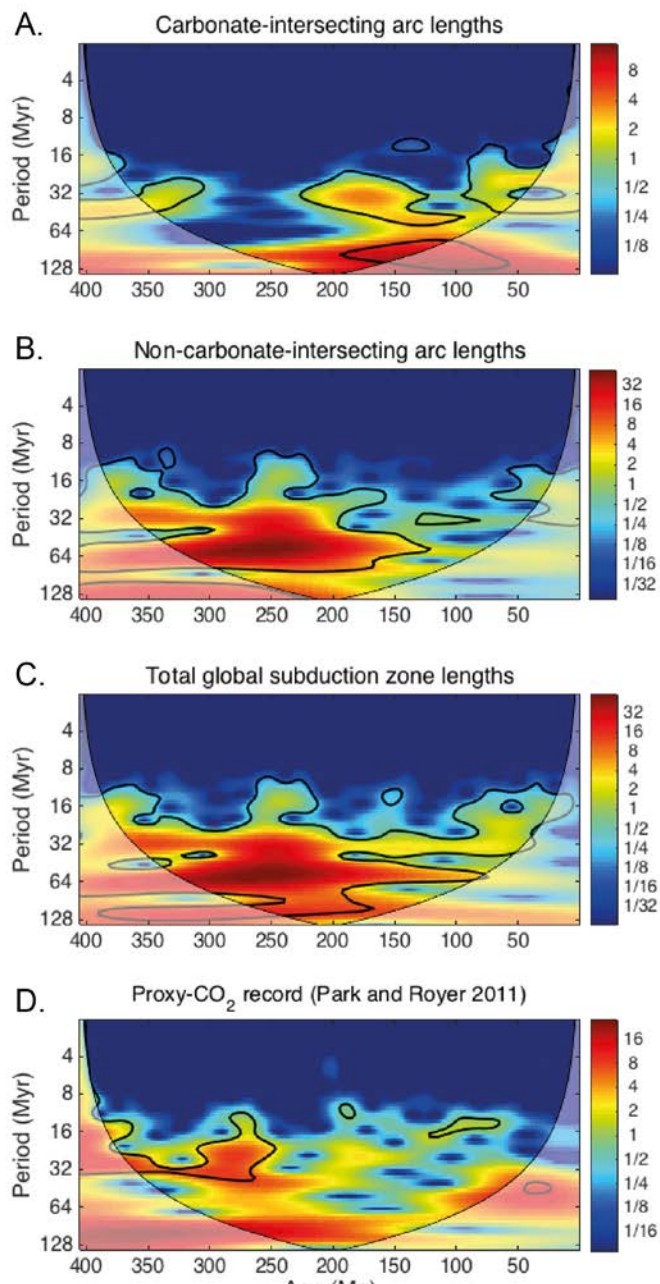

**Figure 2:** Continuous wavelet transforms (CWT) for (a) CIA lengths; (b) NCIA lengths; (c) total global subduction zone lengths and (d) the proxy-$CO_2$ data from Park and Royer (2011). Thick black contours designate the 5% significance level against a red noise background spectrum. The cone of influence (white translucent region) is where edge effects distort the spectrum. The colour bar indicates the wavelet power, with hotter colors corresponding to the maxima. Note the logarithmic scale of the Period (Myr) axis.



## 4.2 Cross-wavelet Transform (XWT) and wavelet coherence (WTC)

The periodic terms common in the arc length time series and the proxy-$CO_2$ record were investigated further using XWT and WTC. Figure 3 shows the XWT comparing the proxy data and the carbonate-intersecting arc lengths data. A distinct lack of co-varying peaks for periods less than ~5 Myrs in length reflects the low-pass filter attenuating small-scale oscillations in all time series. Strong joint power at scales greater than 64 Myrs was detected between both the CIA lengths and NCIA lengths and the proxy-$CO_2$ record, however, the result is never above the 95% confidence level (Fig. 3b).

Intervals of high joint power between CIA lengths and proxy-$CO_2$ trends (410-280 Ma, 210-150 Ma and 75-0 Ma) are interspersed with intervals of low joint power (280-210 Ma and 150-80 Ma). In the ~32 Myr band during 360-280 Ma, peaks in CIA activity precede peaks in palaeo-atmospheric $CO_2$ by close to 90º and in the ~40 Myr band during 75-0 Ma, where the majority of the significant region lies outside the COI (Fig. 3a). In other words, during these intervals CIA activity leads atmospheric $CO_2$ levels by ~ 6-8 Myr.  Only the interval from 75-0 Ma is confirmed by WTC as being coherent through time. In the 210-150 Ma time interval, $CO_2$ peaks exhibit a ~45-90º lead (~5 Myr) over CIA activity to the anti-phase in the 20-40 Myr band (Fig. 3a). Considering the non-uniqueness problem outlined by Grinsted et al. (2004), the region of 90º phase difference may also be interpreted as a ~5 Myr in-phase lag of $CO_2$ after CIA peaks. The lack of significant coherence in this region suggests coincidence rather than causation.





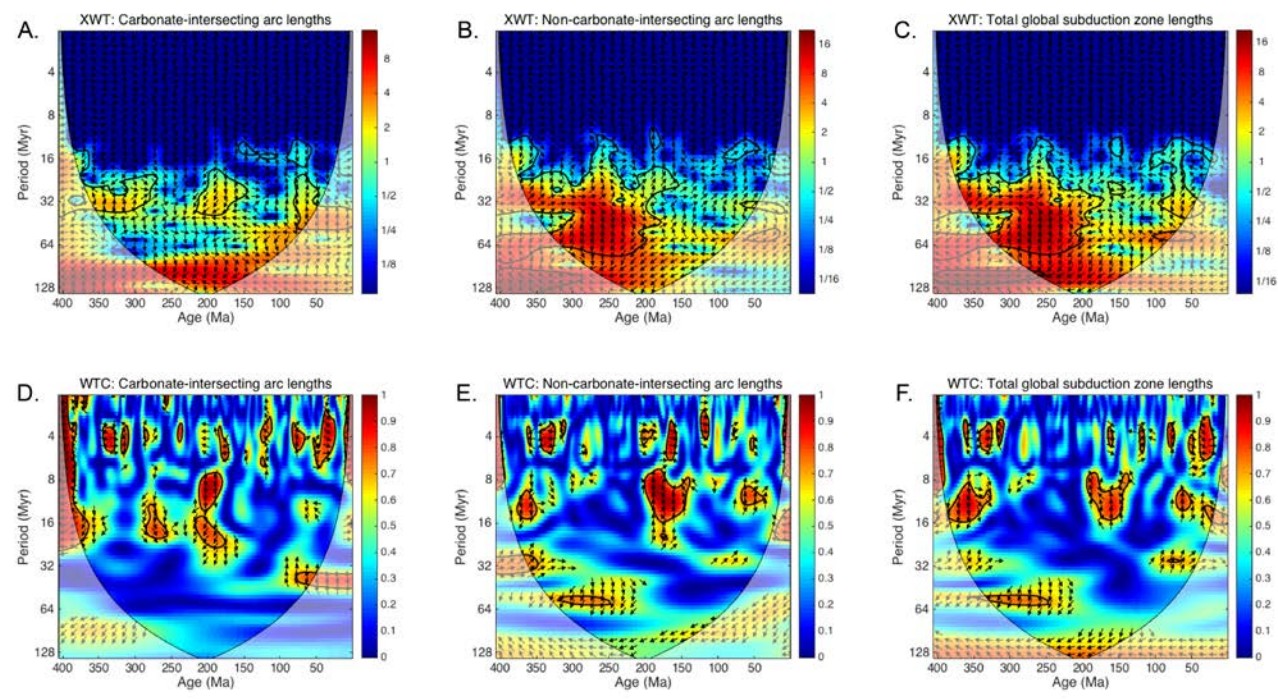

**Figure 3:** The cross-wavelet transforms (XWT) (top) and wavelet coherence (WTC) (bottom) for (a,d) CIA lengths; (b,e) NCIA lengths; (c,f) total global subduction zone lengths respectively and the proxy-$CO_2$ record (Park and Royer 2011). Thick black contours designate the 5% significance level against a red noise spectrum. The white translucent region designates the cone of influence (COI). The colour bar indicates the magnitude of cross-spectral power for the XWT, and for the WTC it represents the significance level of the Monte-Carlo test. The arrows indicate the phase relationship of the two time series in time-frequency space, where east-pointing arrows indicate in-phase behaviour, west-pointing arrows indicate anti-phase behaviour, north-pointing arrows indicate that the arc or subduction zone activity leads the $CO_2$ peaks and south-pointing arrows indicating that $CO_2$ peaks lead arc or subduction zone activity.

The WTC between CIA lengths and the proxy-$CO_2$ record does not show appreciable and prolonged coherence through time (Fig. 3d). However, at distinct intervals the WTC confirms the significance of three regions of high cross-amplitudes: the 14-24 Myr periodicity between 380-360 Ma; the 18-24 Myr periodicity between 200-180 Myr; and the 40-48 Myr periodicity from 75-0 Ma (although much of the significant region lies outside the COI and thus is not localised in time). Phase arrows for periodicities prevalent during 380-360 Ma and 200-180 Ma indicate that CIA length peaks are anti-correlated with $CO_2$ peaks and lag by ~-60° (2-4 Myrs) whereas from 75-0 Ma, locked phase arrows in the 20-40 Myr band indicate arc length peaks lead $CO_2$ by approximately 90° (10 Myrs) (Fig. 3d). Noisy regions of high



coherence and mixed phase signals that coincide with regions of low cross-spectral power for < 8 Myr periodicities are disregarded as artefacts as the low-pass filter attenuated short-wavelength frequencies. Large regions in the 24-40 Myr band of high joint power (i.e. for 410-280 Ma and 210-150 Ma) were not confirmed by wavelet coherence (Figs. 3a, 3d).

NCIA peaks also shares high joint power with proxy-$CO_2$ peaks at similar intervals to CIA peaks in the 20-48 Myr band, but in contrast shows very high joint power between 275-200 Ma where CIA peaks do not (Fig. 3b). In the longer-wavelength band (40-80 Myr) between 300-200 Ma, high joint power exists between NCIA and proxy-$CO_2$ peaks a trend not exhibited in CIA activity (Figs. 3b, 3a). The phase relationship shows the opposite of climate forcing behaviour, where NCIA peaks lag 8-20 Myrs behind

palaeo-atmospheric $CO_2$ peaks.

Only a few significant areas of the XWT between NCIA and proxy-$CO_2$ data is confirmed by wavelet coherence above the 95% confidence level. Some of these areas are common between proxy-$CO_2$ and both CIA and NCIA lengths, specifically at 375-350 Ma and 190-175 Ma (Figs. 3d, 3e). In the 10-15 Myr waveband between 375-350 Ma, anti-phase, negatively-correlated behaviour exists where palaeo-

atmospheric $CO_2$ peaks lead NCIA lengths by approximately 90º (3-4 Myrs) (Fig. 3e). Considering the non-uniqueness problem of phase analysis, this could be interpreted as a positively correlated signal with arc activity leading palaeo-atmospheric $CO_2$. This trend is also exhibited between $CO_2$ peaks and CIA lengths during a similar time interval (Fig. 3d). Significant but short-lived coherence between NCIA lengths and the proxy-$CO_2$ record in the 10-16 Myr and ~18 Myr waveband between 190 and 175 Ma

approximately corresponds to an area of coherent power between CIA and proxy-$CO_2$ data (Figs. 3d, 3e). A notable difference is that phase arrows in the 10-16 Myr waveband indicate that the NCIA and proxy-$CO_2$ peaks are in phase with NCIA peaks leading by between 45º to 90º (3-4 Myr), whereas anti-correlated behaviour exists between CIA length peaks and proxy-$CO_2$, where CIA lengths lag proxy-$CO_2$ peaks by the same degree (45º-90º). The phase-opposite discrepancy can be explained by CIA and NCIA lengths,

which sum to total subduction zone lengths, having directionally opposite trends between 190 and 175 Ma (Fig. 4).





Between 400 and 350 Ma, significant coherence confirms cross-spectral power and closely in-phase behaviour in the 30-32 Myr waveband of NCIA lengths, yet most of the significant region lies outside the COI (Figs. 3b, 3d). Phase arrows in the ~64 Myr band between 310 and 250 Ma suggest palaeo-atmospheric $CO_2$ peaks led NCIA peaks by approximately 10 Myrs, the opposite of climate forcing

5 behaviour. The XWT and WTC for proxy-$CO_2$ correlations with total subduction zone lengths exhibit a very similar spectrum to results for NCIA lengths (Figs. 3e, 3f). One exception is that coherence in the 30-32 Myr waveband becomes dominant between 90-70 Ma, a correlation that was not above the 5% significance level when examining NCIA lengths. Phase relationships in this interval indicate that in-phase $CO_2$ peaks are leading by ~60º (~5 Myrs).

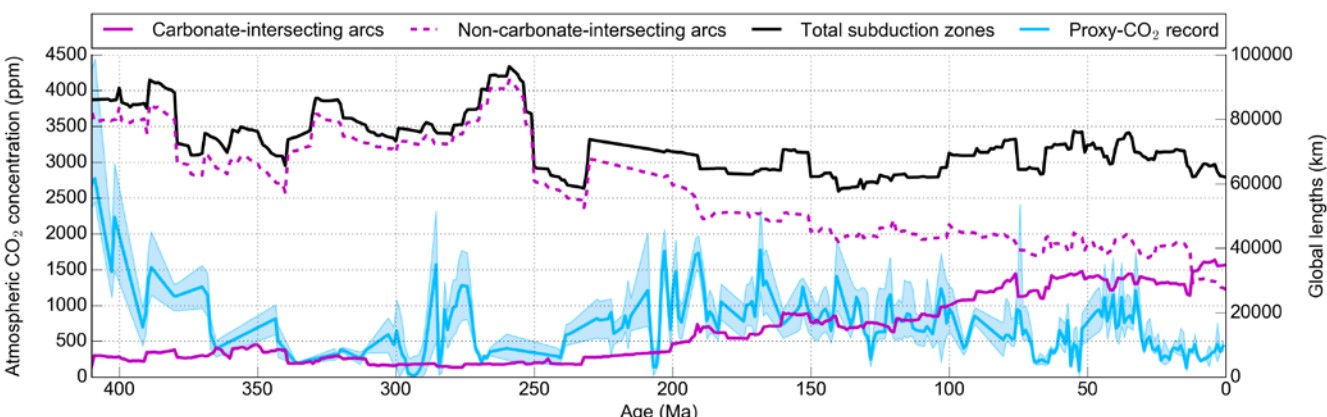

**Figure 4:** Total global lengths of subduction zones (black,) (as a surrogate measure for volcanic arc lengths) compared with arcs that intersect with buried crustal carbonate platforms (magenta, solid) and arcs that do not intersect crustal carbonate platforms (magenta, dashed). The proxy-$CO_2$ record (blue line) from Park and Royer (2011) with upper and lower bounds (blue envelope) are displayed for comparison.

15 **4.3 Comparing carbonate-intersecting arc length and palaeo-$CO_2$ trends**

Coherence analysis suggests that CIA lengths and palaeo-atmospheric $CO_2$ peaks may not be well correlated prior to 100 Ma, however, during some time windows, periodic behaviour may be correlated. These three areas of significance highlighted by the XWT and WTC analysis are between 380-350 Ma, 210-190 Ma and 75-50 Ma.

20 Between 380-350 Ma, coherence analysis highlights possible in-phase 30-32 Myr periodic behaviour between both global subduction zone lengths and NCIA lengths and $CO_2$ peaks which is validated by



trend data despite much of the significant region lies outside the COI (Fig. 3d). Decreasing total subduction zone and NCIA lengths mirror the $CO_2$ record trend as lengths drop from ~92 000 km to ~69 000 km and ~83 000 km to ~57 000 km respectively (Fig. 4), yet the signal was coherent at the 95% significance level for NCIA lengths (Figs. 3e, 3f). In contrast, wavelet analysis found the opposite of

climate forcing behaviour for CIA lengths over this interval where $CO_2$ peaks led CIA lengths (Fig. 3c), and in line with the analysis, CIA lengths remain relatively low (3 000-10 000 km) over this period (Fig. 4). It is evident that the magnitude of change in CIA lengths cannot explain the precipitous drop in palaeo-atmospheric $CO_2$ (2 750 ppm to 200 ppm) taking place from ~410 Ma to ~340 Ma (Fig. 4).

Non-unique interpretations of phase behaviour in the WTC are non-conclusive regarding whether total

subduction zones, CIA or NCIA activity or $CO_2$ peaks are leading during the 210-190 Ma interval. It appears in the XWT and WTC that CIA activity lags palaeo-atmospheric $CO_2$ peaks, whereas NCIA activity and total subduction length peaks lead palaeo-atmospheric $CO_2$ peaks (Figs. 3e, 3f). We observe a rapid reduction in NCIA lengths (~63500 to 50000 km) and global subduction zones (~ 71 000 to 65 000 km) simultaneously with a relatively dramatic increase CIA lengths, which almost double (~7400 to

~16300 km) between 210-190 Ma (Fig. 4). However, as only small-scale periodic behaviour (~10 Myrs) exists for palaeo-$CO_2$ but not for CIA or NCIA lengths during this period point to a scale mismatch in the CWT results (Fig. 2). Hence, neither CIA, NCIA nor global subduction zone lengths can adequately explain large-amplitude (~45-~1 800 ppm), high-frequency (~10 Myr) fluctuations in palaeo-atmospheric $CO_2$ levels over the 210-190 Ma period (Fig. 4).

Between 75-0 Ma, XWT and WTC analysis highlighted a significant domain of high coherence between CIA length data and the proxy-$CO_2$ record in the 40-48 Myr waveband with an arc-leading relationship of ~10 Myrs (Fig. 3c, 3d). Over the same time interval, an area of high coherence with arc-leading relationships occurs in the ~32 Myr waveband between global subduction zone lengths and the proxy-$CO_2$ record (Figs. 3c, 3f). This result is reflected in trend data, where CIA and global subduction zone

lengths reach a local maximum at ~75 Ma (~32 000 km and ~74 000 km respectively), drop sharply for a period of ~10 Myrs before recovering between 63-50 Ma (on average ~32 000 and ~72 000 km in length, respectively) (Fig. 4). The proxy-$CO_2$ data mirror this decline-and-recovery trend, as palaeo-



atmospheric $CO_2$ levels decline substantially between 75-70 Ma (~1200 ppm to ~250 ppm) and recover after 60 Ma (~700 ppm) (Fig. 4). NCIA-$CO_2$ XWT results highlight significant joint power during the same interval, but it was not confirmed as coherent behaviour by the WTC (Fig. 3b, 3d). In accordance, NCIA lengths do not sustain the same linked behaviour as CIA and global subduction lengths over this

interval.

## 5 Discussion

The Accumulation Model was constructed in a way that resulted in increased subduction zone interactions with carbonate platforms in the overriding plate through time. Indeed from 150 Ma onward, CIA lengths consistently made up ~70% of total global subduction zone lengths (Fig. 4). Given this cumulative effect,

we would expect that the growing reservoir of carbonate platforms would result in linked, periodic behaviour in the more recent past compared to 400 Ma if CIAs had a climate-forcing effect. Ultimately, our study does not support the hypothesis that carbonate-intersecting arcs have contributed substantially to atmospheric $CO_2$ levels over the past 410 Myrs. However, potential issues with the large uncertainties in the proxy record, complexities in reconstructing pre-Pangea subduction zones and carbonate platforms,

as well as the requirement to filter the time series may influence this result. CIA lengths and the proxy-$CO_2$ record are largely independent with no strong leader relationship that is consistent through time, with the exception of the most recent ~75 Myrs.

Wavelet analysis highlights many intervals of significant and high joint power in all XWT spectra, yet they were not coherent and thus could not be distinguished from coincidence, such as the region of high

power in ~10-64 Myr wavebands from 300-200 Ma for global subduction zone lengths and NCIA lengths (Figs. 3b, 3c). In some intervals, wavelet analysis highlights the opposite of climate-forcing behaviour between CIA lengths and palaeo-atmospheric $CO_2$ levels (i.e. where $CO_2$ peaks precede arc lengths), such as between 210 and 190 Ma (Fig. 3a). Similarly, a long-lived coherent ~64 Myr signal in both the NCIA lengths and global subduction zone lengths was found to be a significantly coherent signal at the 95%

level (Figs. 3e, 3f), yet the $CO_2$ peak occurred ~16 Myrs prior to NCIA and global subduction length peaks and the result was not found to be meaningful in the context of climate-forcing behaviour (Fig. 4).





A 30-32 Myr periodicity in the NCIA length signal in the 375-350 Ma interval may be interpreted as being linked to a concomitant decline in palaeo-atmospheric $CO_2$, a relationship which was similar in the global subduction zone lengths signal yet was insignificant at the 95% level. Both palaeobotanical and modelling investigations suggest that the rise of vascular land plants (Algeo et al., 1995) and heightened

5    chemical weathering (Berner and Kothavala, 2001) were the dominant forces controlling the large reduction in atmospheric $CO_2$ during the Late Devonian, however, this result does not preclude the notion that waning CIA lengths may have partially contributed to the cooler background climate. Our results suggest that arc lengths have not consistently modulated the $CO_2$-forced transitions between warmer and cooler climates throughout the Phanerozoic. However similar trends in CIA lengths and global subduction

10   zone lengths between 75-50 Ma suggest that the Late Cretaceous to Early Palaeogene warm climate may be a phenomenon forced by global subduction zone lengths rather than carbonate-intersecting arc lengths alone.



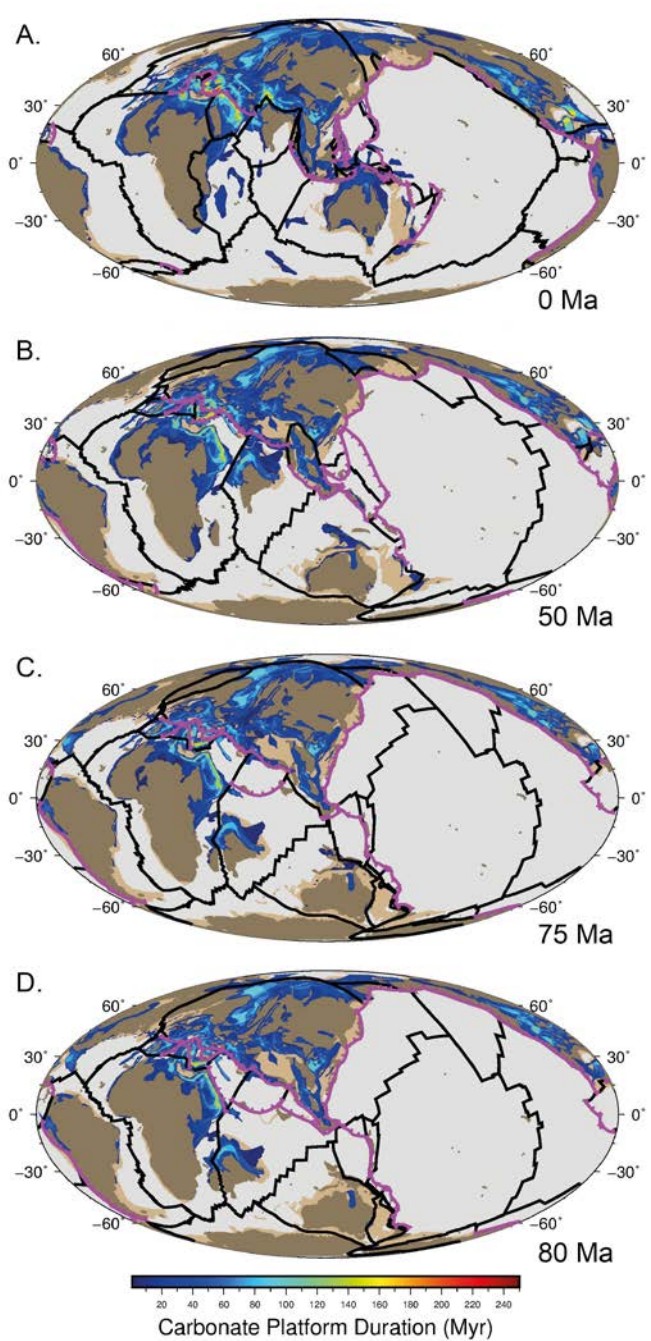

**Figure 5:** Plate reconstructions with plate boundaries (black), subduction zones (purple) and distributions of carbonate platforms. Carbonate platforms are distributed according to the Accumulation Model at 0, 75 and 80 Ma. Colour bar corresponds to the duration of time that the carbonate platforms were actively developing for in the crust.



## 5.1 Implications for Late Cretaceous to Early Palaeogene Climate

Only during the most recent 75 Myrs is it plausible that CIA lengths influenced atmospheric $CO_2$, yet there are two factors that prevent us from viewing these correlations as causative. Firstly, while the ~32 Myr cyclicity is coherent and significant at the 5% level, it is not time-localised between 60-0 Ma as the result lies outside the COI. Secondly, CIA length trends passively mirror total global subduction zone lengths during this time, suggesting that it is not specifically the $CO_2$ liberated at CIAs that drove atmospheric $CO_2$ peaks, but net global volcanic $CO_2$ emissions.

However, the period between 75-50 Ma seems to be distinct from other times as a significant coherency occurs. The changing distribution of CIA lengths and subduction zones between 75 and 50 Ma may have contributed to the well-studied greenhouse climate 51-53 Ma, known as the Early Eocene climatic optimum (EECO), where temperatures were about $14 \pm 3\,°C$ warmer than the pre-industrial period (Caballero and Huber, 2013). While climate interactions and feedbacks leading to the greenhouse state remain uncertain, climate sensitivity to $CO_2$ forcing is likely (Anagnostou et al., 2016).

According to the Accumulation model, the dramatic change that characterised the reduction in CIA lengths ~75 Ma included the termination of a ~3 000-km-long section of a north-dipping intra-oceanic subduction zone in the Northern Tethys ocean, corresponding to the Trans-Tethyan Subduction System (Jagoutz et al., 2016), and the cessation of subduction along the Sundaland margin (McCourt et al., 1996; Zahirovic et al., 2016) (Figs. 5c, 5d). The intra-oceanic subduction in the Neo-Tethys in our model would not be classified as a carbonate-intersecting arc due to the overriding plate (in this case largely oceanic lithosphere) lacking extensive carbonate platforms. In any case, this intra-oceanic subduction zone in the Accumulation Model corresponds to the emplacement of the peri-Arabian ophiolite belt along the Afro-Arabian continental margin between 80-70 Ma rather than an Andean-style arc, the erosion of which has been posited as a cause of global cooling following the EECO (Jagoutz et al., 2016) (Figs. 5c, 5d). We can therefore rule out arc-related volcanism in this region as a contributor to high $CO_2$ levels given that the shut-down of this arc is not associated with the cessation of Andean-style subduction.



The termination of arc volcanism particularly along the Sundaland margin, where a large area of carbonate platforms are buried, contributed ~3 000 km to the global reduction in CIA lengths by ~7 000 km between 75-65 Ma (Figs. 5b, 5c). The resumption of subduction along the Sunda-Java trench at 65 Ma in the model as well as the emplacement of a set of carbonate platforms in the lower latitudes at 63 Ma explain the

modelled results of an increase in CIA lengths at 63 Ma which persist past 50 Ma (Fig. 5b). The EECO occurred approximately 10 Myr later, supporting the 10 Myr lag in peaks of a ~40 Myr periodicity found by the CIA-proxy-$CO_2$ XWT and WTC (Fig. 3a). Our analysis shows a concomitant increase in NCIA lengths (Fig. 4), but the periodic behaviour was not found to be significantly coherent and so coincidental peaks cannot be ruled out. Modern volcanic $CO_2$ output from arcs in Indonesia, Papua New Guinea, parts

of the Andes and Italian Magmatic Province are strongly influenced by carbonate assimilation (Mason et al., 2017) and volcanoes in those regions currently contribute significantly to global atmospheric $CO_2$ flux (Carter and Dasgupta, 2015). Assuming a similar subduction style, this result gives plausibility to the idea that volcanic activity along the Sunda-Java trench during the Cretaceous was also significantly contributing to global $CO_2$ output.

Previous studies have suggested that global arc $CO_2$ contributed to the baseline warm climate of the Late Cretaceous and late Palaeogene, due to the greater size of carbonate reservoirs in continents and the increased continental arc lengths (Lee et al., 2013; Carter and Dasgupta, 2015; McKenzie et al., 2016; Cao et al., 2017). Our results agree well with the hypothesis, as a relative increase in CIA lengths has been linked to a peak in atmospheric $CO_2$ from 75-50 Ma, contributing to enhanced $CO_2$ degassing.

However, it cannot be determined from this study whether the increase in global arc lengths is more important than the relative increase in CIA lengths.

### 5.2 Limitations of wavelet analysis

Wavelet analysis is inherently sensitive to the shape of signals, and as such, adjustment of filtering and processing techniques introduces variations in the results. The choice of interpolation method of the

proxy-$CO_2$ record may subdue actual oscillations in the signal, affecting phase relationships and confidence intervals.  Similarly, a size-7 moving average window filter was chosen to remove high-



frequency noise, yet it may have moved trends forward in time, which ultimately changes whether peaks overlap or not and adds uncertainty to results. Peaks in the XWT were robust in that they were found to appear in XWT plots regardless of the intensity of smoothing, yet the significance contours differed strongly due to smoothing. For instance, with a longer low-bandpass filter, small-scale oscillations

<5 Myrs were removed, reducing areas of significant power in the shorter wavebands. Similarly, when experimenting with a high-bandpass filter, significant regions were directed away from the longer periodicities of >64 Myrs to shorter periodicities. The low-bandpass filter was chosen to remove the noise for smaller periodicities where fluctuations were less likely to represent trends and more likely to represent artefacts of interpolation, yet this measure erases the fidelity of short-term oscillations in the

time series. Hence detail on the <10 Myr scale is not adequately represented, and thus our study was unable to evaluate short-term relationships. While sensitivity testing was carried out on filtering, interpolation and smoothing techniques of the proxy-$CO_2$ record, a robust sensitivity analysis was beyond the scope of this study. Additionally, the nature of using a tectonic framework to estimate total subduction zone lengths does not lend well to estimations of error as only one model was used to extract arc length

data, and so only one signal could be extracted. In this way, the selection and comparison of two signals with large uncertainty are not sufficient in eliciting periodic behaviour and are not likely to demonstrate true correlations.

XWT analysis describes areas with high power which can be misleading when the XWT spectrum is not normalised. For instance, areas of high joint power were found where relatively flat peaks in the CIA

length data occurred simply because of the large amplitude of peaks in the proxy-$CO_2$ data. The vast differences in magnitude do not necessarily constitute a causative relationship. WTC analysis does not present this problem because the spectrum is normalised, and hence periodicities with strong power were not necessarily found to be coherent (Maraun and Kurths, 2004). The combination of both measures somewhat circumvents the problem of misleading peaks in the XWT, but not all. Given that wavelet

analysis is very sensitive to magnitude and timing of peaks, the fact that the uncertainty in our modelled estimates of arc lengths are not well-constrained introduces variability in correlative relationships.



Wavelet analysis, like other signal processing techniques, is limited by the length of the time series data investigated. Periods of greater than half the time series length cannot be examined using this method (Winder, 2002). Therefore, very-long oscillations on the scale of 250 Myr or more could not be derived from our data. One reason for examining patterns on 250 Ma periods is due to the very-long-term nature

of supercontinent cycles. The accretion of supercontinents such as Pangea and Rodinia are associated with shortened continental arc lengths, whilst continental break-up and dispersal initiates subduction on the edges of continents, and thus increases continental arc lengths (Donnadieu et al., 2004; Lee et al., 2016). Not only do changing modes of supercontinent break-up or accretion have the potential to influence climate, the opposite causal effect may occur where changes in the long-term climate state can

feed back into the coupled system and initiated changes in the tectonic regime (Lenardic et al., 2016). Given that supercontinent cycles exceed 200 Myr in length and exert a large effect on the length of continental arcs (Lee et al., 2013), applying this analysis to plate reconstructions that capture multiple supercontinent cycles (E.g. Merdith et al., 2017) to investigate whether such long-term linkages exist between atmospheric $CO_2$ and global arc volcanism.

## 5.3 Limitations of model assumptions

The extent to which our models are representative of volcanic arc $CO_2$ emissions is limited by some issues with the subduction zone quantification approach. The subduction zone modelling attempted to approximate relative magmatic $CO_2$ flux without directly measuring it by assuming a constant $CO_2$ flux and a unit-correspondence of arc lengths to volcanoes. Both assumptions are problematic. There are

several dynamic factors that govern arc $CO_2$ flux such as convergence rates (van der Meer et al., 2014), the composition and volume of subducted sediments (Rea and Ruff, 1996), convergence obliquity (Kerrick and Connolly, 2001), slab volume flux (Fischer, 2008), the relative contribution of metamorphic decarbonation in the crust and in the mantle wedge (Keleman and Manning, 2015) and the efficiency of decarbonation (Johnston et al., 2011). The model does not account for these factors and assumes that the

parameters of $CO_2$ flux have remained constant in time. In addition, the number of volcanoes per unit length of subduction zones is neither constant through time nor correlated, at least for the circum-Pacific





arc and Central American arc (Kerrick, 2001). Furthermore, the Accumulation Model has no way to account for carbonate platform thicknesses, and thus cannot account for the realistic depletion of crustal carbonate reservoirs through time. The inability of the model to incorporate the complexity of depleting carbonate platforms may lead to sampling bias during continental collisions. This highlights an area of

future improvement. Taking all limitations into consideration, the Accumulation Model still provides a first-order approximation of changing arc lengths through time as a surrogate measure for $CO_2$ degassing, especially for the early Phanerozoic which had not previously been modelled at a 1 Myr resolution.

**6 Conclusions**

Wavelet analysis is a useful tool in elucidating coherent, phase-related periodicities between the $CO_2$

record and various climate forcing factors. Our analysis has revealed that CIA activity is largely independent to the proxy-$CO_2$ record over the past 410 Myr except for the period 75-50 Ma. The CIA lengths that we derived from an Accumulation model of carbonate platform evolution do not find reasonable and persistent periodicities that explain atmospheric $CO_2$ flux, however there may be some causal relationships in the Late Cretaceous to Early Palaeogene that are supported by previous modelling

efforts.

This analysis lends partial support to the idea that, at certain times, carbonate-intersecting arc activity is more important than non-carbonate intersecting activity in driving atmospheric $CO_2$, yet the climate system is vastly complex and the result could mean that other processes have dominated climate feedback mechanisms through time, such as the proliferation of terrestrial plant life (Algeo et al., 1995), silicate

weathering (Kent and Muttoni, 2008) and the emplacement of large igneous provinces (Belcher and Mander, 2012).

Generally, an absence of constraints on carbon-climate interactions in deep-time makes it difficult to test hypotheses about greenhouse and icehouse climate states. Clearly, more accurate reconstructions of climate-forcing processes are needed to improve our understanding of the deep carbon system. It is

suggested that further investigations be carried out at specific time intervals to eliminate the uncertainty around the time periods within the COI at the end of the data series (between 410-350 Ma and 50-0 Ma).





With adjustments to the parameters and assumptions governing the carbonate platform evolution models, the data can be used as input into fully-coupled planetary-scale climate models. This would provide the most comprehensive and self-consistent approach to understand the contribution of global arc volcanism to the deep carbon cycle.





## Code and data availability

The subduction zone toolkit by Doss et al. (2016) was developed as a product of this research and has been made publicly available for use with open-source plate reconstruction software, GPlates. The toolkit includes all proxy $CO_2$ data, pyGPlates and bash scripts as well as the Matthews et al. (2016) plate model.

5  Both the Signal Processing Toolbox™ and Wavelet Coherence Toolbox™ for MATLAB® were used in analysis and the production of figures. The Wavelet Coherence Toolbox™ can be found at noc.ac.uk/marine-data-products/cross-wavelet-wavelet-coherence-toolbox-matlab, while the Signal Processing Toolbox™ can be found at au.mathworks.com/help/signal/index.html.

10  Doss, S., Zahirovic, S., Müller, D. and Pall, J: DCO Modelling of Deep Time Atmospheric Carbon Flux from Subduction One Interactions: Plate Models & Minor Edits, Zenodo, http://dx.doi/org/10.5281/zenodo.154001, 2016.

## Author Contribution

Sebastiano Doss and Jodie Pall developed the Subduction Zone toolkit together and carried out modelling experiments under the supervision of Dietmar Müller and Sabin Zahirovic. Figures were prepared by Jodie Pall and Sebastiano Doss, and the toolkit repository was prepared by Sebastiano Doss. Jodie Pall

conducted wavelet analysis with assistance from Rakib Hassan. Jodie Pall also prepared the manuscript with contributions from all co-authors.

## Competing interests

The authors declare that they have no conflict of interest.

## Acknowledgements

The authors acknowledge and thank the Deep Carbon Observatory (DCO) for funding this deep carbon modelling and visualisation effort. We would also like to thank members of the EarthByte Group for all their kind help as well as the University of Sydney for supporting open source research.

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
