# Peer review of "The influence of carbonate platform interactions with subduction zone volcanism on palaeo-atmospheric CO2 since the Devonian"

_Climate of the Past, 2017_

## Referee Comment (RC1) · Anonymous Referee #1 · 1 Nov 2017

General comments:

This is a clear, well-written analysis of an important problem in paleoclimate: what is the connection between tectonic activity and the carbon cycle/climate? The hypothesis is clearly stated and the use of plate reconstructions for the analysis is a useful and appropriate way to approach the problem.

The wavelet analysis appears to be rigorous. However, I wonder why this method was chosen rather than, for example, a simpler autocorrelation? Autocorrelation would appear to address the same hypothesis without assuming periodic behavior. In fact, the existence of periodic signals on the order of 10s of millions of years is very surprising –

if those signals are real and significant, then the authors should explore possible mechanisms for generating such periodic changes in CO2 and/or arc activity (e.g. around pg 9, Line 17-18 and pg 11, section 4.1). It is not clear to me how or why periodic signals should appear from tectonic interactions with carbonate platform. Without a proposed mechanism, perhaps the signals are simply noise in the data?

I also find it very surprising that arcs interacting with carbonate platforms seem to have increased 5-fold (as shown in Fig. 4) from 250 to 50 Ma. Is this result perhaps an artifact of only mapping out Phanerozoic platforms from the Kiessling 2003 database? Why are Precambrian platform areas not included? There are many examples of known, extensive Precambrian carbonate platforms, and I suspect that adding them to the analysis would remove a significant portion of this signal. It seems unlikely that such a major change in magnitude could occur based on tectonic interaction with a depositional environment known to have existed since the Archean. If this result is robust, then the authors should advance some possibilities for why this dramatic change occurred.

If the above comments can be addressed, this study demonstrates a useful application of global plate reconstructions for examining Earth system behavior over the last ∼400 Myr.

Specific comments:

One aspect of the analysis was unclear to me: are the locations of past arcs mapped out in ancient plate reconstructions? This was the impression I received from the description of the plate reconstruction model, the mapping of carbonate platforms, and Fig. 5. However, this impression seemed to be contradicted by Fig. 4 caption and pg 4, lines 6-8, which describe using subduction zone lengths as a surrogate for volcanic arc lengths. Why is the latter necessary if the arcs and plate boundaries can be accurately mapped out? I hope the authors can clarify their methods.

Fig 2/3: If filtering has removed any signal < ∼5Myr, that portion of the results should

be removed (or at least indicated).

Perhaps refer to Figure S1 (showing proxy CO2 data) when those data are mentioned (pg 9, line 24 and pg 10, line 10). Additionally - how is noise/uncertainty in the proxy data accounted for? How is the sparseness of data older than 220 Ma addressed?

Technical corrections: Pg 7, line 11: 'temporally limited to the Devonian' is unclear (it sounds like only the Devonian is being analyzed). Consider specifying that the maximum time considered in the analysis is the Devonian. Pg 9, line 4: wording is awkward: 'corresponds to an upper limit by which carbonate platforms can interact...', consider changing 'by which' to 'for interactions of carbonate platforms...' Pg 10, line 22: Wording: 'modelled data' is unclear.

————————————————————

---

## Author Comment (AC1) · 11 Nov 2017

Manuscript Title: Arc volcanism, carbonate platform evolution and palaeo-atmospheric CO2: Components and interactions in the deep carbon cycle

Manuscript Number: cp-2017-112

General Comments by Anonymous Referee #1:

Referee comment: This is a clear, well-written analysis of an important problem in paleoclimate: what is the connection between tectonic activity and the carbon cycle/climate? The hypothesis is clearly stated and the use of plate reconstructions for

the analysis is a useful and appropriate way to approach the problem. The wavelet analysis appears to be rigorous. However, I wonder why this method was chosen rather than, for example, a simpler autocorrelation? Autocorrelation would appear to address the same hypothesis without assuming periodic behavior.

Author response: Autocorrelation calculates the correlation of a signal with a delayed copy of itself. In other words, it analyses periodicity in signals in a single time series and cannot be applied to compare two time series like those in our study. The equivalent of autocorrelation for two time series is cross-correlation, which is the time-domain equivalent of cross-spectral analysis in the Fourier domain. However, neither cross-spectral correlation nor cross-correlation can characterise correlation as a function of time or scale, unlike in wavelet analysis. Wavelet analysis, which is similar to cross-spectral correlation, is fundamentally different as it does not use sine or cosine waves to characterise a signal, but (as in our case) Morlet wavelets. It then deconvolves the signal into constituent Morlet wavelets to attempt to find similarities between two time series, irrespective of whether periodicities exist in signals. In this way, wavelet transforms are not only used for detecting periodicities in signals but to detect any time- or space-dependent correlations in signals. For clarity, justification of the wavelet analysis method were added on pg. 3, line 17-24 and pg. 4, line 4-6.

Referee comment: In fact, the existence of periodic signals on the order of 10s of millions of years is very surprising – if those signals are real and significant, then the authors should explore possible mechanisms for generating such periodic changes in CO2 and/or arc activity (e.g. around pg 9, Line 17-18 and pg 11, section 4.1).

Author response: The referee is correct. However, in our analysis there were no periodicities < ~32 Myr that were found to be significant and meaningful. Instead, only short-term in-phase behaviour has been investigated in the discussion which links a peak in carbonate-intersecting arc (CIA) lengths to a peak in palaeo-atmospheric CO2. In agreement with the referee's comments, the possible mechanisms for arc-related periodicities have been indicated in the introduction (see pg. 3, line 12-15), however the

authors believe that further explanation of the mechanisms is not warranted given the results, and is beyond the scope of the paper.

Referee comment: It is not clear to me how or why periodic signals should appear from tectonic interactions with carbonate platform. Without a proposed mechanism, perhaps the signals are simply noise in the data?

Author response: Following on from response #1, wavelet analysis was applied to find regions of correlation and not necessarily periodicities. However, it can be applied to uncover periodicities if there are any. Hence, the objective of this paper is to find correlations and periodicities that have not yet been explored. Lee et al. (2013) identified the possibility of periodicity which is a key assumption in the paper. Inclusion of this assumption has been added to the Introduction on pg. 3 line 14-17. Patterns of CO2 storage occur when island arcs dominate continental arcs and CO2 liberation when continental arcs dominate, which are linked to the periodic assembly and dispersal of continents. Earlier work by Lendardic et al. (2011) propose a mantle thermal mixing mechanism to explain supercontinent cycles that alter the nature of subductions in ways that effect CO2 release. Our investigation assumed that mechanisms such as those explored by Lenardic et al. (2011) and Lenardic (2016) and hypothesised by Lee et al. (2013) exist and cause periodic linked behaviour between arc activity and atmospheric CO2. However, we do not focus on the mechanisms as it would be beyond the scope of the paper. Secondly, the use of the cross-wavelet spectrum with wavelet coherence is applied to make sure that signals found in the data are real and significantly different from simple noise.

Referee comment: I also find it very surprising that arcs interacting with carbonate platforms seem to have increased 5-fold (as shown in Fig. 4) from 250 to 50 Ma. Is this result perhaps an artifact of only mapping out Phanerozoic platforms from the Kiessling 2003 database?

Author response: The result the referee highlights is because an Accumulation Model

of carbonate platforms was used which assumes that crustal carbonate reservoirs grow in thickness and spatial extent through time. Unless there is a dramatic change in arc regimes (i.e. from continental to intraoceanic), CIA lengths will only increase. The limitations of this assumption are discussed in section 2.2.

Referee comment: Why are Precambrian platform areas not included? There are many examples of known, extensive Precambrian carbonate platforms, and I suspect that adding them to the analysis would remove a significant portion of this signal5. It seems unlikely that such a major change in magnitude could occur based on tectonic interaction with a depositional environment known to have existed since the Archean. If this result is robust, then the authors should advance some possibilities for why this dramatic change occurred.

Author response: It is true that the Kiessling et al. (2003) compilation of global, palaeo-distribution of carbonate platform maps extend to the Ordovician, however incorporating these now would require an extensive amount of work. The reason why no platforms earlier than the Devonian were mapped was because initially an 'Accumulation Model' and an 'Active Carbonate Platform Model' were tested and compared. The Active Carbonate Platform model assumed that crustal carbonate reservoirs could be depleted over tens of millions of years, and entailed that platform existed within certain windows. For the results to be comparable, carbonate platform evolution started at the same time. Given that our plate model only records plate motions from 410 Ma, both models were only implemented from the Devonian. However, modelling carbonate platforms from the earliest Phanerozoic and Precambrian is part of future work. In any case, the Cone of Influence (COI) means that the earliest and latest part of the time series are affected by distortion, such that our wavelet analysis excludes characterising the Devonian (and late Cretaceous to present day). As such, we believe that adding the Ordovician and Silurian platforms would not significantly impact wavelet analysis, which is the focus of our study.

Referee comment: If the above comments can be addressed, this study demonstrates

a useful application of global plate reconstructions for examining Earth system behavior over the last âĹij400 Myr.

Specific Comments by Anonymous Referee #1: Referee comment: One aspect of the analysis was unclear to me: are the locations of past arcs mapped out in ancient plate reconstructions? This was the impression I received from the description of the plate reconstruction model, the mapping of carbonate platforms, and Fig. 5. However, this impression seemed to be contradicted by Fig. 4 caption and pg 4, lines 6-8, which describe using subduction zone lengths as a surrogate for volcanic arc lengths. Why is the latter necessary if the arcs and plate boundaries can be accurately mapped out? I hope the authors can clarify their methods.

Author response:We can only map out plate boundaries and, hence, subduction zone lengths using our analysis. This is why we used subduction zones as a surrogate for continental arc lengths. However, to avoid confusion, adjustments have been made in Section 2.3 (pg. 7, line 27; pg. 8 line 1) and Section 2.4 (pg. 8, line 10) and the caption of Figure 4 (pg. 16).

Referee comment: Fig 2/3: If filtering has removed any signal < âĹij5Myr, that portion of the results should be removed (or at least indicated).

Author response: Results referring to short wavelength (<5 Myr) periodicity have been removed from: pg. 13, line 4-5; pg. 14, line 17-18; pg. 15, line 17-18; and pg. 16, line 1.

Referee comment: Perhaps refer to Figure S1 (showing proxy CO2 data) when those data are mentioned (pg 9, line 24 and pg 10, line 10).

Author response: The referee's suggestion has been implemented with corrections on pg. 9, line 22 and pg. 10, line 12.

Referee comment: Additionally - how is noise/uncertainty in the proxy data accounted for? How is the sparseness of data older than 220 Ma addressed?

[Figure]

Author response: Noise and uncertainty in the proxy data cannot be accounted for using our approach, nor can our plate model, as one working hypothesis, account for uncertainty. However the authors recognise the importance of quantifying error and uncertainty, and have highlighted this as an area of future work. A paragraph that addresses error quantification has been added to the discussion (pg. 24, line 8-15).

Technical corrections: Referee comment: Pg 7, line 11: 'temporally limited to the Devonian' is unclear (it sounds like only the Devonian is being analyzed). Consider specifying that the maxi-mum time considered in the analysis is the Devonian10.

Author response:'Temporally limited to the Devonian' was changed to 'in the Phanerozoic from the Devonian to present day' on pg. 7, line 10.

Referee comment: Pg 9, line 4: wording is awkward: 'corresponds to an upper limit by which carbonate platforms can interact...' consider changing 'by which' to 'for interactions of carbonate platforms...'

Author response: The referee's suggestion to change 'by which' to 'for interactions of carbonate platforms. . .' was accepted on pg. 8, line 27; pg. 9, line 1.

Referee comment: Pg 10, line 22: Wording: 'modelled data' is unclear12.

Author response: 'Modelled data' was changed to 'estimates from the Accumulation model' on pg. 10, line 21.

Author comments:

Additional corrections were made in the text, which are described below: 1. The colour of the subduction zone in Figure 1 has been updated to reflect actual colours of the figure, and text has been updated on the figure. 2. 'One-to-one' changed to general on pg. 4, line 18. 3. 'Continuously' removed from pg. 5, line 5. 4. 'Colors' changed to 'colours' for consistency with British English in Figure 2. 5. A short paragraph was moved based on order of figures (pg. 11, line 19-21). 6. Text has been removed that is not considered to add anything to analysis from pg. 11, line 10-12 and pg. 16, line

18-26. 7. 'E.g.' changed to 'e.g. on pg. 25, line 5

---

## Author Comment (AC2) · 11 Nov 2017

**Author's Response to Anonymous Referee #1**

Manuscript Number: cp-2017-112

Manuscript Title: Arc volcanism, carbonate platform evolution and palaeo-atmospheric CO2: Components and interactions in the deep carbon cycle

**General Comments by Anonymous Referee #1:**
This is a clear, well-written analysis of an important problem in paleoclimate: what is the connection
between tectonic activity and the carbon cycle/climate? The hypothesis is clearly stated and the use of plate reconstructions for the analysis is a useful and appropriate way to approach the problem.
The wavelet analysis appears to be rigorous. However, I wonder why this method was chosen rather than, for example, a simpler autocorrelation? Autocorrelation would appear to address the same hypothesis without assuming periodic behavior.

*Authors' response: Autocorrelation calculates the correlation of a signal with a delayed copy of itself. In other words, it analyses periodicity in signals in a single time series and cannot be applied to compare two time series like those in our study. The equivalent of autocorrelation for two time series is cross-correlation, which is the time-domain equivalent of cross-spectral analysis in the Fourier*
*domain. However, neither cross-spectral correlation nor cross-correlation can characterise correlation as a function of time or scale, unlike in wavelet analysis. Wavelet analysis, which is similar to cross-spectral correlation, is fundamentally different as it does not use sine or cosine waves to characterise a signal, but (as in our case) Morlet wavelets. It then deconvolves the signal into constituent Morlet wavelets to attempt to find similarities between two time series, irrespective of*
*whether periodicities exist in signals. In this way, wavelet transforms are not only used for detecting periodicities in signals but to detect any time- or space-dependent correlations in signals. For clarity, justification of the wavelet analysis method were added on pg. 3, line 17-24 and pg. 4, line 4-6.*

In fact, the existence of periodic signals on the order of 10s of millions of years is very surprising – if those signals are real and significant, then the authors should explore possible mechanisms for generating such periodic changes in CO2 and/or arc activity (e.g. around pg 9, Line 17-18 and pg 11, section 4.1).

*The referee is correct. However, in our analysis there were no periodicities < ~32 Myr that were found to be significant and meaningful. Instead, only short-term in-phase behaviour has been investigated in the discussion which links a peak in carbonate-intersecting arc (CIA) lengths to a peak in palaeo-atmospheric CO₂. In agreement with the referee's comments, the possible mechanisms for arc-related periodicities have been indicated in the introduction (see pg. 3, line 12-*
*15), however the authors believe that further explanation of the mechanisms is not warranted given the results, and is beyond the scope of the paper.*

i

It is not clear to me how or why periodic signals should appear from tectonic interactions with carbonate platform. Without a proposed mechanism, perhaps the signals are simply noise in the data?

*Following on from response #1, wavelet analysis was applied to find regions of correlation and not necessarily periodicities. However, it can be applied to uncover periodicities if there are any. Hence, the objective of this paper is to find correlations and periodicities that have not yet been explored. Lee et al. (2013) identified the possibility of periodicity which is a key assumption in the paper. Inclusion of this assumption has been added to the Introduction on pg. 3 line 14-17. Patterns of $CO_2$ storage*
*occur when island arcs dominate continental arcs and $CO_2$ liberation when continental arcs dominate, which are linked to the periodic assembly and dispersal of continents. Earlier work by Lendardic et al. (2011) propose a mantle thermal mixing mechanism to explain supercontinent cycles that alter the nature of subductions in ways that effect $CO_2$ release. Our investigation assumed that mechanisms such as those explored by Lenardic et al. (2011) and Lenardic (2016) and hypothesised*
*by Lee et al. (2013) exist and cause periodic linked behaviour between arc activity and atmospheric $CO_2$. However, we do not focus on the mechanisms as it would be beyond the scope of the paper.*

*Secondly, the use of the cross-wavelet spectrum with wavelet coherence is applied to make sure that signals found in the data are real and significantly different from simple noise.*

I also find it very surprising that arcs interacting with carbonate platforms seem to have increased 5-fold (as shown in Fig. 4) from 250 to 50 Ma. Is this result perhaps an artifact of only mapping out Phanerozoic platforms from the Kiessling 2003 database?

*The result the referee highlights is because an Accumulation Model of carbonate platforms was used*
*which assumes that crustal carbonate reservoirs grow in thickness and spatial extent through time. Unless there is a dramatic change in arc regimes (i.e. from continental to intraoceanic), CIA lengths will only increase. The limitations of this assumption are discussed in section 2.2.*

Why are Precambrian platform areas not included? There are many examples of known, extensive Precambrian carbonate platforms, and I suspect that adding them to the analysis would remove a significant portion of this signal[5]. It seems unlikely that such a major change in magnitude could occur based on tectonic interaction with a depositional environment known to have existed since the Archean. If this result is robust, then the authors should advance some possibilities for why this dramatic change
occurred.

*It is true that the Kiessling et al. (2003) compilation of global, palaeo-distribution of carbonate platform maps extend to the Ordovician, however incorporating these now would require an extensive amount of work. The reason why no platforms earlier than the Devonian were mapped was*
*because initially an 'Accumulation Model' and an 'Active Carbonate Platform Model' were tested*

*and compared. The Active Carbonate Platform model assumed that crustal carbonate reservoirs could be depleted over tens of millions of years, and entailed that platform existed within certain windows. For the results to be comparable, carbonate platform evolution started at the same time. Given that our plate model only records plate motions from 410 Ma, both models were only*
*implemented from the Devonian. However, modelling carbonate platforms from the earliest Phanerozoic and Precambrian is part of future work.*

*In any case, the Cone of Influence (COI) means that the earliest and latest part of the time series are affected by distortion, such that our wavelet analysis excludes characterising the Devonian (and late Cretaceous to present day). As such, we believe that adding the Ordovician and Silurian platforms*
*would not significantly impact wavelet analysis, which is the focus of our study.*

If the above comments can be addressed, this study demonstrates a useful application of global plate reconstructions for examining Earth system behavior over the last ∼400 Myr.

**Specific Comments by Anonymous Referee #1:**
One aspect of the analysis was unclear to me: are the locations of past arcs mapped out in ancient plate reconstructions? This was the impression I received from the description of the plate reconstruction model, the mapping of carbonate platforms, and Fig. 5. However, this impression seemed to be contradicted by Fig. 4 caption and pg 4, lines 6-8, which describe using subduction zone lengths as a
surrogate for volcanic arc lengths. Why is the latter necessary if the arcs and plate boundaries can be accurately mapped out? I hope the authors can clarify their methods.

*We can only map out plate boundaries and, hence, subduction zone lengths using our analysis. This is why we used subduction zones as a surrogate for continental arc lengths. However, to avoid*
*confusion, adjustments have been made in Section 2.3 (pg. 7, line 27; pg. 8 line 1) and Section 2.4 (pg. 8, line 10) and the caption of Figure 4 (pg. 16).*

Fig 2/3: If filtering has removed any signal < ∼5Myr, that portion of the results should be removed (or at least indicated).
*Results referring to short wavelength (<5 Myr) periodicity have been removed from: pg. 13, line 4-5; pg. 14, line 17-18; pg. 15, line 17-18; and pg. 16, line 1.*

Perhaps refer to Figure S1 (showing proxy CO2 data) when those data are mentioned (pg 9, line 24 and
pg 10, line 10).

*The referee's suggestion has been implemented with corrections on pg. 9, line 22 and pg. 10, line 12.*

Additionally - how is noise/uncertainty in the proxy data accounted for? How is the sparseness of data
older than 220 Ma addressed?

*Noise and uncertainty in the proxy data cannot be accounted for using our approach, nor can our plate model, as one working hypothesis, account for uncertainty. However the authors recognise the importance of quantifying error and uncertainty, and have highlighted this as an area of future work. A paragraph that addresses error quantification has been added to the discussion (pg. 24, line 8-15).*

Technical corrections: Pg 7, line 11: 'temporally limited to the Devonian' is unclear (it sounds like only the Devonian is being analyzed). Consider specifying that the maxi-mum time considered in the analysis is the Devonian[10].

*'Temporally limited to the Devonian' was changed to 'in the Phanerozoic from the Devonian to present day' on pg. 7, line 10.*

Pg 9, line 4: wording is awkward: 'corresponds to an upper limit by which carbonate platforms can interact...' consider changing 'by which' to 'for interactions of carbonate platforms...'

*The referee's suggestion to change 'by which' to 'for interactions of carbonate platforms…' was accepted on pg. 8, line 27; pg. 9, line 1.*

Pg 10, line 22: Wording: 'modelled data' is unclear[12].

*'Modelled data' was changed to 'estimates from the Accumulation model' on pg. 10, line 21.*

**Author comments:**

Additional corrections were made in the text, which are described below:
- The colour of the subduction zone in Figure 1 has been updated to reflect actual colours of the figure, and text has been updated on the figure.
- 'One-to-one' changed to general on pg. 4, line 18.
- 'Continuously' removed from pg. 5, line 5.
- 'Colors' changed to 'colours' for consistency with British English in Figure 2.
- A short paragraph was moved based on order of figures (pg. 11, line 19-21).
- Text has been removed that is not considered to add anything to analysis from pg. 11, line 10-12 and pg. 16, line 18-26.
- 'E.g.' changed to 'e.g. on pg. 25, line 5

[revised manuscript text omitted]

---

## Referee Comment (RC2) · Anonymous Referee #2 · 13 Nov 2017

This is an interesting study that explores links between arc activity and the long-term CO2 budget. This is not a new effort, as the authors acknowledge, but the authors apply a statistically rigorous method that most reliably tests the relationships between two signals, subduction zone length and atmospheric CO2. This manuscript is also very well written and constructed. That said, I do have a few concerns on the data compilation, methodology, and the interpretations. I do not have a good grasp of wavelet analyses used in this paper, so I would not comment on the methodology. However, I understand that the conclusions are based on the reconstruction and data compilations.

[Figure]

The study is partially inspired by the hypothesis in Lee et al. papers. I notice that "continental arc" in Lee et al. is not equivalent to "CIA" in this study, because thickened crusts lead to evolved magma that is more efficient in wall rock decarbonation when intruding.

It is good that the authors did not extract CO2 signals from models like GEOCARB, so did not mix model results with the data extracted from natural samples. In addition to the uncertainty issues noted by the authors, GEOCARB scales tectonic input of CO2 to the spreading rate of mid-ocean ridges, so does not explicitly account for magmatic-metamorphic outgassing at arcs. Although faster MOR spreading in general corresponds to more vigorous global tectonics, arc activities also depend on the configuration of plates. Thus, it is logically inconsistent to compare arc lengths with GEOCARB model results. The authors apply a filter to remove high-frequency noises (note: "noise" might not perfectly appropriate in this context), but I doubt whether the resolution and uncertainties in ages of Park and Royer data are within a few million years. I suggest that the authors briefly summarize and discuss both uncertainties and temporal resolution of the data compilation of Park and Royer (2011).

I have a major concern on the accumulation model used in this study. Early Paleozoic and pre-Cambrian carbonate platform deposits are entirely ignored in the reconstruction, so the CIA curve (Fig. 4) starts at 0 km at 410 MA, which is unrealistic. The CIA lengths almost monotonically increase, and the present CIA length is about 6 times of Permian and 1.5-1.2 times of Cretaceous. This curve alone, without any wavelet analyses, would falsify the hypothesis of Lee et al., misleading the readers to conclude that the contribution of arc activity to Earth's long-term climate is minimal. As the authors have noted, the accumulation model provides an upper bound because it is assumed that the platform carbonate was not depleted in geologic events (erosion, subduction etc.). It is only fair to compare the arc lengths with CO2 proxies if the authors also provide a lower bound estimate.

The CIA and NCIA lengths are derived from the GPlates reconstruction and 24 palaeographic maps. It is not clear to me how the authors differentiate arc lengths from lengths of convergent margins, or subduction versus collision zones. Again, it will help the readers to assess the quality of reconstruction if uncertainties are discussed in addition to the model assumptions and limitations. In the current version it sounds a bit like almost no error! The authors need to justify that the data and model reach a 1-Myr resolution (Line 7, Page 26). The present global length of CIA in Figure 4 is about 35,000 km, more than double of the lengths of continental arcs measured from geologic maps (15,000 km; Fig. 4d in Cao et al., 2017 EPSL). This large discrepancy makes me worry about the GPlates reconstruction in this study. Where does the extra CIA length account for? It is essential to address this discrepancy in the revised manuscript for the readers to understand the meaning of CIA defined in this study.

I am extremely curious to know how wavelet analysis using the compilation of Cao et al. (2017) or McKenzie et al. (2017) and the $CO_2$ record (Park and Royer, 2011) would turn out.

It is not clear to me how arcs on subducted fossil plates are constructed in GPlates or whether this portion of fossil arc (subduction zone) is added in Fig. 4. This is not directly relevant to the conclusion of this study, but the authors should state the assumptions and protect their model results from being misinterpreted.

I don't think it is a good idea to mix all decarbonation processes at convergent margins in Section 5.3. These are not the limitations of THIS model that addresses the relationship between arcs and long-term $CO_2$. Instead, the authors might focus on a series of assumptions and limitations in the reconstruction and data compilation (comments above).

To address these concerns, it potentially requires substantial work of model development and data compilation and it seems that this will take long. That is why I suggest rejection but with strong encouragement to resubmit. I hope the authors are willing to perform a major revision, as it will significantly strengthen their arguments. I would like

to say that I have great respect for the work that has been done in this project and for this research group in general, but I cannot be positive at this time. I very much hope that my comments help to improve the manuscript.
* * *

---

## Author Response (AR1)

**Author's Response**

Manuscript Number: cp-2017-112

5  Manuscript Title: Arc volcanism, carbonate platform evolution and paleo-atmospheric CO2:
Components and interactions in the deep carbon cycle

Dear Laurie Menviel,

10  Thank you for highlighting those grammatical and spelling mistakes. In this copy I have made all
suggested changes, which were:

*L16, p2: "at" MORs*
*Figure 1 legend: (a) and (b) are reversed*
15  *L. 5, 10 and 11 on p10: Please move the https addresses to "Code and data availability".*
*L. 10 p14L "are" interspersed*
*L. 16 p 15: by "an" average*
*Figure 4 legend: Please correct the colour description of the lines (for example total length of
subduction is black in graph but dashed magenta in legend).*
20  *L. 7 p 22: "suggesting"*

Warm Regards,

25  Jodie Pall

[revised manuscript text omitted]

20 162, 293-337, 2016.

---

## Author Response (AR2)

**Author's Response to Anonymous Referee #2**

Manuscript Number: cp-2017-112

Manuscript Title: Arc volcanism, carbonate platform evolution and palaeo-atmospheric CO2: Components and interactions in the deep carbon cycle

**General Comments by Anonymous Referee #2:**

Referee comment:
This is an interesting study that explores links between arc activity and the long-term CO2 budget. This is not a new effort, as the authors acknowledge, but the authors apply a statistically rigorous method that most reliably tests the relationships between two signals, subduction zone length and atmospheric CO2. This manuscript is also very well written and constructed. That said, I do have a few concerns on the data compilation, methodology, and the interpretations. I do not have a good grasp of wavelet analyses used in this paper, so I would not comment on the methodology. However, I understand that the conclusions are based on the reconstruction and data compilations.

The study is partially inspired by the hypothesis in Lee et al. papers. I notice that "continental arc" in Lee et al. is not equivalent to "CIA" in this study, because thickened crusts lead to evolved magma that is more efficient in wall rock decarbonation when intruding.

Author response:
*We originally differentiate the 'continental arc' in the Lee et al. (2013) and Cao et al. (2017) papers from our work with the term 'carbonate intersecting arc [CIA],' which was defined to capture the entire contribution of decarbonation and the release of $CO_2$ into the atmosphere along subduction zones. As Cao et al. (2017) are quantifying only volcanic $CO_2$ along continental arcs, to call our subduction zone measurements a surrogate for 'arc activity' is no longer appropriate. This is because our estimations of subduction zone lengths capture a greater source of $CO_2$ than arc volcanism alone, including diffuse degassing from crustal magmatic processes. As such, we have redefined the terms 'carbonate-intersecting continental [CIC] subduction zones' and 'non-carbonate-intersecting continental [non-CIC] subduction zones,' which are complementary and together make up global subduction zones (pg. 3, lines 11-13).*

Referee comment:
It is good that the authors did not extract CO2 signals from models like GEOCARB, so did not mix model results with the data extracted from natural samples. In addition to the uncertainty issues noted by the authors, GEOCARB scales tectonic input of CO2 to the spreading rate of mid-ocean ridges, so does not explicitly account for magmatic-metamorphic outgassing at arcs. Although faster MOR spreading in general corresponds to more vigorous global tectonics, arc activities also depend on the configuration of plates. Thus, it is logically inconsistent to compare arc lengths with GEOCARB model results. The authors apply a filter to remove high-frequency noises (note: "noise" might not perfectly

appropriate in this context), but I doubt whether the resolution and uncertainties in ages of Park and Royer data are within a few million years. I suggest that the authors briefly summarize and discuss both uncertainties and temporal resolution of the data compilation of Park and Royer (2011).

5    Author response:
*In accordance with the referee's suggestion, the uncertainties in the Park and Royer (2011) compilation were summarised (pg. 23, lines 24-27).*

Referee comment:
10   I have a major concern on the accumulation model used in this study. Early Paleozoic and pre-Cambrian carbonate platform deposits are entirely ignored in the reconstruction, so the CIA curve (Fig. 4) starts at 0 km at 410 MA, which is unrealistic. The CIA lengths almost monotonically increase, and the present CIA length is about 6 times of Permian and 1.5-1.2 times of Cretaceous. This curve alone, without any wavelet analyses, would falsify the hypothesis of Lee et al., misleading the readers to conclude that the
15   contribution of arc activity to Earth's long-term climate is minimal.

Author response:
*We agree with the referee regarding the construction of the Accumulation Model. It was originally designed to exhibit the evolution of carbonate platforms to cover the timeframe of the plate tectonic*
20   *reconstructions, and so only included all platforms back to 410 Ma. This has been corrected in the current model, which includes Ordovician and Silurian platforms as hypothesised by Kiessling et al. (2003). Reference is made to the creation of this model with the earlier Paleozoic platforms (pg. 5, lines 16, 22-24). We would like to note that the CIC subduction zone curve (previously referred to as the CIA curve) did not start from 0 km, but from 3140 km. It is noted that in the new model, CIC*
25   *subduction zone lengths are estimated to be 7940 km in length at 410 Ma (pg. 17, line 8-9).*

Referee comment:
As the authors have noted, the accumulation model provides an upper bound because it is assumed that the platform carbonate was not depleted in geologic events (erosion, subduction etc.). It is only fair to
30   compare the arc lengths with CO2 proxies if the authors also provide a lower bound estimate.

Author response:
*To create a lower bound estimate would mean creating a model where one can reasonably assume that carbonate platforms have been depleted over time. We attempted a model like this, where*
35   *carbonate platforms were actively interacting with subduction zones only in the geological period within which they were produced, based on the Kiessling et al. (2003) compilation (Figs. R1, R2). This lower bound estimate model assumed that older, buried carbonate platforms were depleted or 'inactive', and were not interacting with arc magmatism. The results indicated that the spatial area of carbonate platforms in this model was consistently lower than the Accumulation Model (Fig. R1),*
40   *and implied that carbonate platforms produced in the past are no longer significant reservoirs in the present (Fig. R2). We found these results and the assumption of fast depletion to be unrealistic. This is especially true because the carbonate platforms mapped by Kiessling et al. (2003) are currently in*

[Figure]

*existence, and it is not known how carbonate platform carbon contents have depleted through time. While we agree that a lower bound estimate is useful, it is not possible to create one for the past 410 Ma with the data set we are working with. Instead, we state that we are providing a first-pass approximation based on a set of assumptions that are clearly defined.*

[Figure]

Fig R1. Total global lengths of subduction zones (black) compared with CIC subduction zones (magenta) which is included in our manuscript, and our lower bound estimate of non-accumulating CIC subduction zone lengths (red).

[Figure]

Fig R2. Plate reconstructions with plate boundaries (black), subduction zones (purple) and distribution of carbonate platforms in the Accumulation model (upper) and in the lower bound estimate model (lower). The colour bar corresponds to the duration of time that the carbonate platforms were actively developing in the crust.

Referee comment:
The CIA and NCIA lengths are derived from the GPlates reconstruction and 24 palaeo graphic maps. It is not clear to me how the authors differentiate arc lengths from lengths of convergent margins, or subduction versus collision zones. Again, it will help the readers to assess the quality of reconstruction if uncertainties are discussed in addition to the model assumptions and limitations. In the current version it sounds a bit like almost no error!

iv

Author response:

*We attempt to make our analysis clearer by redefining lengths of 'carbonate-intersecting continental [CIC] subduction zones' and 'non-CIC subduction zones,' (pg. 4, lines 4-7). We are not solely concerned with volcanic $CO_2$ emissions but rather the combination passive (diffuse) and active $CO_2$ emissions along all subduction zones, and so we believe that we no longer need to differentiate the types of convergent margins.*

Referee comment:
The authors need to justify that the data and model reach a 1-Myr resolution (Line 7, Page 26).

Author response:
*While we cannot find where the referee is pointing to on page 26, reference to the 1 Myr resolution of our model is made on pg. 4, lines 20-21. The key components are carbonate platforms that are assumed to be active over a segment of geological time in the Kiessling et al. (2003) compilation. These are linked to plate reconstructions that provide evolving plate boundaries in 1 Myr intervals, thus allowing us to track the subduction zone and carbonate platform interactions at finer temporal resolutions than the original Kiessling et al. (2003) snapshots.*

Referee comment:
The present global length of CIA in Figure 4 is about 35,000 km, more than double of the lengths of continental arcs measured from geologic maps (15,000 km; Fig. 4d in Cao et al., 2017 EPSL). This large discrepancy makes me worry about the GPlates reconstruction in this study. Where does the extra CIA length account for? It is essential to address this discrepancy in the revised manuscript for the readers to understand the meaning of CIA defined in this study.

Author response:
*The Cao et al. (2017) approach is very different to our approach in the way that it compiled arc volcanics from active margins. For plate tectonic reconstructions, one can have a subduction zone and produce very little arc volcanism (e.g., southern Turkey, parts of the Andean margin, etc. – see figure below), while we know that tectonic convergence and subduction is occurring. That subduction is carrying volatiles into the mantle wedge, with partial melting likely being trapped in the crust (rather than being extruded) to produce more diffuse CO2 emissions. In essence, the Cao et al. (2017) work provides a minimum length of continental volcanic arcs (largely because it relies on comprehensive sampling and preservation of volcanic rocks), while our work represents a likely upper bound on continental arc lengths.*

[Figure]

Fig R3. Present-day subduction zones (red) from Bird et al. (2003), compared with Holocene volcanism related to subduction (purple) or other tectonic settings (yellow). The active (bright green) and buried (light green) carbonate platforms from Kiessling et al. (2003) are also plotted.

[Figure]

Fig R4. Present-day continental arcs (red lines) from Cao et al. (2017) [left] and present-day oceanic and continental subduction zones (magenta lines) from the GPlates reconstructions used in this study [right]. The obvious differences include that we treat the Mediterranean (e.g. parts of Italy, etc.) and much of east Asia (e.g. Japan) as continental arcs, unlike in Cao et al. (2017). Intra-oceanic subduction zones are not used in our study.

Referee comment:

I am extremely curious to know how wavelet analysis using the compilation of Cao et al. (2017) or McKenzie et al. (2017) and the CO2 record (Park and Royer, 2011) would turn out. It is not clear to me how arcs on subducted fossil plates are constructed in GPlates or whether this portion of fossil arc (subduction zone) is added in Fig. 4. This is not directly relevant to the conclusion of this study, but the authors should state the assumptions and protect their model results from being misinterpreted.

Author response:

*Applying the analysis to the Cao et al. (2017) material is a significant undertaking, largely because the subduction zones are plotted on single snapshots (representing longer geological timeframes) on an entirely different plate reconstruction. This is outside the scope of this work, but should be a component of future work.*

Referee comment:

I don't think it is a good idea to mix all decarbonation processes at convergent margins in Section 5.3. These are not the limitations of THIS model that addresses the relationship between arcs and long-term CO2. Instead, the authors might focus on a series of assumptions and limitations in the reconstruction and data compilation (comments above).

Author response:

*As the referee mentions, there is very little work that has been done on disambiguating the decarbonation processes at subduction zones. While some modelling work has been undertaken (e.g. Gonzales et al., 2016) it is difficult to distinguish the contribution of crust from subducting sediments, and the relative contribution of other factors including the angle and thickness of the subducting slab. As such, we are forced to make a simplistic assumption that subduction zone lengths are correlated with $CO_2$ emissions in our analysis, and subsequently make our assumptions and limitations clear to the readers (Section 2.2, pg. 7-8; Section 5.3, pg. 25-26 ).*

Referee comment:

To address these concerns, it potentially requires substantial work of model development and data compilation and it seems that this will take long. That is why I suggest rejection but with strong encouragement to resubmit. I hope the authors are willing to perform a major revision, as it will significantly strengthen their arguments. I would like to say that I have great respect for the work that has been done in this project and for this research group in general, but I cannot be positive at this time. I very much hope that my comments help to improve the manuscript.

General comments by the authors:

1.  All instances of CIA and NCIA lengths have been changed to CIC and non-CIC subduction zone

    lengths respectively.

2. Basic assumptions about the $CO_2$ emissions at subduction zones have been redefined and stated (pg. 1, line 3-7; pg. 3, line 9-11, 27-28; pg. 4, line 1-2, section 2, pg. 8, Section 2.4; ).

3. As per suggestion by Referee #2 and the Editor, the inclusion of Ordovician and Silurian Carbonate Platforms has been made to the Accumulation Model, which has been stated in the manuscript (pg. 5, Section 2.1).

4. Assumptions and limitations of the Accumulation Model (pg. 6-7, Section 2.2; pg. 24-25, Section 5.3) and wavelet analysis (pg. 22, line 24-27) have been re-written to better describe the limitations of our single-estimate model.

5. Results, discussion and figures have been updated to reflect new analyses (pg. 11-22).

Author response:

*General comments by the referee have all been accepted and adapted into the text. Further changes to the manuscript include:*

1. *Conducting our analysis again and replacing all figures in the text (Figs. 1-5 and S1-S2) and re-writing the results and discussion to reflect new and different results (Section 4).*

2. *Changing the title to change the focus from volcanic arcs to the broader interactions between active margins and crustal carbonate platforms.*

3. *Changing all instances of 'CIA lengths' to 'CIC subduction zone lengths,' including within figures and captions.*

4. *Specifying why it is difficult to test the hypothesis of Lee et al. (2013) (pg.3, lines 20-21).*

5. *Streamlining text for word economy and comprehension, e.g. pg. 4, line 9.*

6. *Elaborating on the work by Cao et al. (2017) in comparison to our work (pg. 4, lines 9-15).*

7. *Explaining our plate reconstruction in more detail (pg.5, lines 2-14), and our process of measuring subduction zone lengths (pg. 9, lines 6-15).*

[revised manuscript text omitted]

---

## Author Response (AR3)

Manuscript Number: cp-2017-112

Manuscript Title: The influence of carbonate platform interactions with subduction zone volcanism on palaeo-atmospheric CO$_2$ and the deep carbon cycle since the Devonian

Authors: Jodie Pall, Sabin Zahirovic, Sebastiano Doss, Rakib Hassan, Kara J Matthews, John Cannon, Michael Gurnis, Louis Moresi, Adrian Lenardic and R Dietmar Muller.

**General comments by the Editor:**

Editor comment: Although both Reviewers suggest to publish the manuscript with minor revisions, the Reviewers raise two major issues that should be addressed. Both Reviewers are concerned with the omission of Precambrian platforms. The second Reviewer is also concerned by the use of CIC and carbonate-intersecting subduction zones in the text.

The feedback provided in the reviewer assessments of your manuscript is important and should be taken into account as you complete your revision. I encourage you to submit a suitably revised version of your manuscript.

*Authors' response: We appreciate the feedback provided by the Editor, and have endeavoured to address all issues presented by both reviewers. Due to the concerns of Referee #1 and #2, we have included in our Accumulation model Precambrian and early Phanerozoic carbonate occurrences as data points sourced from entries of 'reefs' and 'carbonate lithologies' in the PaleoReefs database and Paleobiology Database, as well as well-studied carbonates in North China (Meng et al., 2011). We approximate the spatial extent of Precambrian platforms with a 200 km buffer radius around each data point, and by creating polygons of carbonate platforms from the Meng et al. (2011) study. We experimented with using a larger buffer, but the 250 km buffer was too large and occasionally captured oceanic crust, and values less than 200 km were likely to be too small. While this is not an exhaustive compilation of all Precambrian carbonates, creating such a compilation project is beyond the scope of our study.*

*Results in Figure R1 below show very little difference between results when Precambrian platforms are included. This is partly because we do not map the entire extent of Precambrian platforms, and because Precambrian platforms occur in areas already accounted for by other platforms through time.*

*Secondly, as Referee #2 highlighted, we have re-defined our terminology to clarify what we are measuring (subduction zone intersections with carbonate platforms) in order to make our work suitable for comparison with other similar works that use different proxies. We have also changed the title of our manuscript to reflect the changes in terminology.*

*As a newer, up-to-date multi-proxy CO₂ record by Foster et al. (2017) has been released since submitting this paper for publication, we have updated our analysis to include this new information. Therefore, we no longer need to interpolate and re-sample the proxy-CO₂ data (recorded in Section 3.1) because the dataset provided by Foster et al. (2017) provides CO₂ estimates at each 1 Myr*
*interval.*

*The manuscript we present here describes a new analysis with an Accumulation model that includes early Phanerozoic and Precambrian carbonate platforms, a new proxy-CO₂ record and changes to the terminology (from carbonate-intersecting continental [CIC] subduction zones to carbonate-*
*intersecting subduction zones [CISZs]). We hope that our efforts satisfy the Editor and Reviewers.*

[Figure]

- Carbonate-intersecting subduction zone (CISZ) lengths (without Precambrian CPs)
- CISZ lengths (including Precambrian CPs)
- Global subduction zone lengths
- CO₂ proxy record (Foster et al., 2017)

Figure R1. Comparison of our analysis of carbonate-intersecting subduction zone (CISZ) lengths with
an Accumulation model that includes Precambrian platforms (red) compared with CISZ lengths from our model that omits Precambrian platforms (purple). Global lengths of subduction zones (black) and the proxy-CO2 record (blue) from Foster et al. (2017) also appear.

**General Comments by Anonymous Referee #1:**
Referee comment: The authors have made many appropriate revisions to this manuscript. I appreciate the detailed discussion of the uncertainties and limitations of the datasets and analysis, and of the importance of publishing what is mostly a 'negative result', guiding future work on this topic. I am still
concerned with the omission of Precambrian carbonate platforms to the dataset. For example, if the remaining crust, not mapped out by the Kiessling 2003 dataset, is some percentage ancient (pre-Devonian) carbonate platform, then some fraction of the non-CIC subduction zones in the analysis would in fact be CIC zones. This would effectively 'dampen' or decrease the magnitude of the effect derived from CIC vs non-CIC areas in the time interval studied. Perhaps the authors can briefly address this possibility or address this with a simple probabilistic parameter (and possibly showing the effect on Fig 4).

*Authors' response: In our latest analysis we have included Precambrian platforms sourced from databases as well as from a comprehensive study of the spatial extent of Precambrian platforms*
*(Meng et al., 2011). We acknowledge the concerns of Referee #1 that the effect of carbonate-intersecting subduction zones are dampened to a degree when we do not account for all Precambrian platforms, and we address the shortcomings of our analysis in Section 2.2 (pg. 8, lines 15-30).*

**Minor comments by Anonymous Referee #1:**
RC: Note that the silicate weathering feedback (or possibly another negative feedback) must operate on approx. million year timescales in order to maintain stable climate, so the conflict between silicate weathering and arc magmatism/tectonics (e.g. pg 2, Lines 22-25) is not really a true opposition. The question is really how tectonics or arcs modify the relationship between the carbon cycle and climate, rather than a choice between two different controls.
*AR: We agree with the referee and have removed lines 22-25 from pg. 2 as we found the information provides, as the referee states, a complicated comparison.*

RC: Pg 3, Line 21: specify wavelet analysis of what variables?
*AR: The variables subject to wavelet analysis in this paper have been specified more clearly on pg. 3,*
*lines 17-22.*

RC: Pg 23, Line 2: size-7 refers to what units? Myrs?
*AR: The size of the average moving window filter has been changed from 7 Myr to 5 Myr in order to smooth out 5 Myr variations and nothing greater. The size of the filter (5 Myr) has been specified on*
*pg. 22, line 26.*

RC: Pg 23, Line 12: the conclusion that detail on <10 Myr timescales is not well represented follows from the coarse resolution of the CO2 proxy data?
*AR: Text on pg. 23, lines 8-9 has been edited to clarify that our study does not focus on geologically*
*short-term oscillations in climate or arc activity.*

RC: Pg 23, Line 19-22: this was unclear to me.
*AR: The paragraph has been rewritten for clarity (pg. 23, lines 13-23).*

**General Comments by Anonymous Referee #2:**
RC: (1) It seems the authors are using CIC and carbonate-intersecting subduction zones (arcs) interchangeably in the text, which can be two different things. What is the exact definition of
continental subduction zones and CIC in this paper?
*AR: We define carbonate-intersecting subduction zones as areas where any subduction zone intersects carbonate platforms buried in the overriding continental crust. We state in Section 2.2 that*

*carbonate reservoirs are more likely to be preserved in continental crust where they are subject less to removal by subduction, which led us to originally conclude that carbonate-intersecting subduction zone lengths would primarily be on continents. However, the referee correctly highlights that in our model we are measuring all subduction zones that intersect with carbonate platforms. At the*

*referee's suggestion, we have changed all instances of carbonate-intersecting continental subduction zones (CIC) to be replaced with carbonate intersecting subduction zones (CISZ). We have also included a more precise definition of CIC and non-CIC subduction zones on pg. 3, lines 24-27.*

RC: I am confused to see the subduction zones (purple curves) in European Alps during 50-0 Ma in
Figure 5. Is this classified as "continental subduction zone" or "continental arc" in this study? Did Andean-type continental arcs exist in European Alps since 50 Ma?
*AR: The convergence between Africa and Europe has been accommodated to a large extent by subduction, preceding the continental collision to build the Alps. The Tyrrhenian Sea opened as a back-arc, in an overall convergent setting, and that system is captured reasonably well by our*
*reconstructions. The evolution of the Carpathians is more complex, but we implement slab rollback, which is supported by a large number of studies (Linzer, 1996; Stampfli and Hochard, 2009). However, there is some debate as to the nature of the subducting slab, and whether it is a lithospheric drip rather than a subducting plate (Royden and Faccenna, 2018).*

RC: The magenta curves in Figure 5 are subduction zones, what are relationships between subduction zones, CIC, non-CIC, oceanic subduction zone (oceanic arc), continental subduction zone (collision belt)? Does non-CIC also include oceanic subduction zone?
*AR: We appreciate that our definitions may be a point of confusion, so have clarified the differences between CI subduction zones and non-CI subduction zones in the Introduction (pg. 3, lines 24-27).*

RC: In the caption of Figure 4, the solid magenta curve is the "all subduction zones that intersect with carbonate platforms." So are you actually dealing with carbonate-intersecting subduction zones instead of carbonate-intersecting continental subduction zones? I am very confused here. Also, can you use another color to highlight the CIC in Figure 5?
*AR: The caption of Figure 4 has been re-written to explain our results with more clarity. While we support the author's suggestion, the CISZs cannot be highlighted in Figure 5 as significant changes in the workflow are required, which the authors were unable to do in the time given for this resubmission.*

RC: Page 18, Line 23: "Our investigation encompasses the total segment length of all subduction zones." So CIC is "all subduction zones that intersect with carbonate platforms" not limiting to continental subduction zones? If this is the case, you may want to clarify/change your terminology and definitions. And, the conclusions about the correlation between CIC and CO2 have to be changed since it is the "all subduction zones that intersect with carbonate platforms" being studied.
*AR: We accept the referee's suggestion and have changed our terminology and explicitly stated our new definitions as outlined previously.*

RC: (2) Only Phanerozoic Carbonate platforms are used. So how about pre-Cambrian ones? The authors mentioned this issue in Section 2.2, but they did not clearly state how such implementation affect their conclusions: would this overestimate or underestimate the length of CIC?

*AR: We have included all instances of reef and carbonate lithologies from the Paleobiology Database and the PaleoReef Database for the Precambrian. From these databases we obtained point-data, and we added a buffer of 200 km to represent some of the spatial extent of these ancient reservoirs. However, mapping the entire spatial extent of Precambrian platforms is beyond the scope of this paper. We have more clearly stated how our conclusions may be influenced by not including*

*Precambrian platforms (pg. 8, lines 23-28).*

RC: Page 8, Line 8, "it is expected that accounting for Precambrian platforms would not drastically change the analysis" Why?

*AR: We believe that accounting for the Precambrian platforms would not significantly change the*

*analysis if the extent of Precambrian platforms largely overlay platforms already accounted for in our model during younger times. We do not consider the thickness of crustal carbonate reservoirs, so our analysis of CISZs would appear very similar. We have updated the text to reflect this.*

RC: For example, during the Proterozoic, extensive carbon platform exists in northern margins of North

China and Tarim plates. In particular, ~ 1.4 Ga Jixian Formation contains about several-km-thick carbonate (e.g., Meng et al., 2011, Stratigraphic and sedimentary records of the rift to drift evolution of the northern North China craton at the Paleo-to Mesoproterozoic transition. Gondwana Research). These carbonates are within 450 km (if use the scheme proposed in this paper) from the northern margin of North China and likely to be intruded by arc magma during Permian-Triassic when Solonker ocean between Mongolia and North China closed.

*AR: As per the referee's suggestion, carbonate platforms from Meng et al. (2011) have been incorporated into our analysis by digitizing the platform polygons and setting them to occur at the first time-frame of our Accumulation model (410 Ma). In Figure R1, the results of our carbonate-intersecting subduction zone lengths with these, and other, Precambrian platforms are mapped*

*against our CISZ lengths without Precambrian platforms. There is very little difference between the two time series, but we nevertheless agree that the Precambrian platforms described by Meng et al. (2011) were useful additions to our Accumulation model.*

RC: (3) Please provide more information on how to read XWT, CWT, WTC figures. For example, in

Figure 2, the area in the cone of influence is less reliable? What is a "red noise background spectrum"? What does "wavelet power" mean? The Results section 4.1-4.2 used many technical terms. For example, "signals of mid- to long-wavelength component remain strong", "waveband", "wavelength", "wavelength band", "periodic component", "peak", what do they mean? Can you translate some of them into the geological language?

*AR: To make sure results can be interpreted by people not familiar to wavelet analysis, we have included in the supplement the following definition of the background power spectrum and have included a definition of wavelet power in the results (pg. 12 lines 4-8 ).*

*The red noise background spectrum is essentially a smoothing process used to increase the confidence of the wavelet spectrum. Red noise is an autocorrelation equation that is characterised by increasing power at lower frequencies, as shown in the spectrum plot of red noise in Figure R2. In the XWT, CWT and WTC, theoretical red noise wavelet power spectra are derived and compared to Monte Carlo results, and is used to establish a null hypothesis for the significance of a peak in the wavelet power spectrum. As stated in the supplements, a red noise process was chosen over a white noise process because it represents geophysical time series better than a white noise spectrum.*

*We have rewritten the results and discussion in an attempt to make the technical terms easier to understand in the manuscript (pg. 12, lines 4-8; pg. 14, lines 3-10, and throughout Sections 4 and 5).*

[Figure]

Figure R2. Periodogram of a red noise model (red) fitted to maximum temperature data (blue) (Image from Timofeyeve and Livezeky, 2018).

**Minor comments by Anonymous Referee #2:**

Referee comment: Page 3, Line 18: Each arc may display 50 Myr flare-ups, but not all of them had flare-ups at the same time (Krisch et al., 2016, American Mineralogist). So the overall global carbon release pattern from these arcs could have the longer temporal wavelength.

*Authors' response: The referee raises a good point, which is that 50 Myr flare-ups may occur at different times along an arc. The text in this manuscript has been updated to better summarise this idea (pg. 4, lines 1-4). Exploring these variations, however, is beyond the scope of our study.*

Page 4, Line 8: Cao et al. (2017)'s compilation of lengths and spatial distributions of continental arcs is based on regional geology and not directly depends on the paleomaps. The history of continental arcs is continuous (see Cao et al.'s Figure 2) and is not based on discrete plate reconstruction. The paleomaps in the Cao et al. (2017) only show snapshots of some specific geological periods.

*Author response: We thank the referee for bringing this to our attention, and we have updated text to more accurately reflect the work that Cao et al. (2017) produced (pg. 4, lines 1-2).*

Page 4, Line 9: Cao et al. (2017) used surface exposures of granitoids as one of the proxies of continental arcs. They are not arc "volcanic" products. They also (1) compensated older arcs with more surface areas of granitoids; (2) used geology to restore the lengths of arcs, which is independent of surface exposures of granitoids; (3) the results from two approaches correlated well in the first order. Given the above points, why does the Cao et al., (2017) represent the lower-limit? It is fine to argue their compilation may miss some continental arcs or underestimate lengths or durations of some arcs. If your length of CIC is carbonate-interesting subduction zones not limiting to continental arcs, you could argue Cao et al., (2017) may underestimate the length of carbonate-interesting arcs.

*AR: We accept the arguments made by the referee regarding the methodology of Cao et al. (2017) and have updated the text to articulate that the Cao et al. (2017) paper may underestimate the flux from carbonate-intersecting arcs due to an under-representation of arcs (pg. 4, lines, 3-7). As an example in Figure R3, at 170 Ma in our study, subduction zones along the Eurasia, eastern Gondwanaland and the eastern Asian margin are longer than in the Cao et al. (2017) study, which is a kinematic requirement to accommodate plate convergence along these margins. As a consequence, we likely capture greater lengths of carbonate-intersecting arcs that are not accounted for in the Cao et al. (2017) model.*

[Figure]

Figure R3. Spatial distribution at 170 Ma of subduction zones and carbonate platforms at in this study (upper) compared to distribution of continental arcs from Cao et al. (2017) (lower).

RC: Page 9, Line 15: why not just using the average trench-arc distance of 287 km? Wondering if the longer distance tends to smooth the temporal pattern. Will the results be affected if 287 km is used instead of 448 km?

*AR: The way we constructed the Boolean carbonate grid mask is to capture any intersections within the highest probability region, i.e. within 1 standard deviation of the mean, as well as any intersections that directly over-lie the subduction zone. If there is a subduction zone interaction with a carbonate platform at any point within the 448 km trench-arc distance along the whisker, then an*

*intersection will be registered at that whisker. Reducing the trench-arc distance will capture only carbonate intersections at the mean trench-arc distance or less, rather than all carbonate platform within one standard deviation of the mean trench-arc distance. We have updated the caption of figure 4 to make this logic easier to understand.*

RC: Page 11, Line 1: Why using 7 Myr (not 6 or 8) window to remove high-frequency signals?
*AR: We used a 7 Myr window because that created a smoother signal. However, as we only want to smooth out 5 Myr oscillations, we have now used a 5 Myr moving average window in our latest analysis.*

RC: Page 11, Line 4: Why do you use 5 Myr here instead of keeping the window size (7 Myr) consistent?
*AR: As above, we now keep the moving average filter of the same window size as the oscillations we are removing from the dominant signal (i.e. 5 Myr).*

RC: Page 18: Line 23: you mean " less than half"?
*AR: We thank the reviewer for noticing this, and the text has been changed to correct this error (pg. 18, line 6).*

[revised manuscript text omitted]

---

## Author Response (AR4)

Manuscript Number: cp-2017-112

Manuscript Title: The influence of carbonate platform interactions with subduction zone volcanism on palaeo-atmospheric $CO_2$ since the Devonian

Authors: Jodie Pall, Sabin Zahirovic, Sebastiano Doss, Rakib Hassan, Kara J Matthews, John Cannon, Michael Gurnis, Louis Moresi, Adrian Lenardic and R Dietmar Muller.

**General comments by the Editor**

Editor's comment: Title: The new title is now (very) long… would it make sense to stop at "CO2"?

*Authors' response: The title has been changed to 'The influence of carbonate platform interactions with subduction zone volcanism on palaeo-atmospheric $CO_2$ since the Devonian.' We believe that the last three words are necessary for contextualising the paper, but agree that 'and the deep carbon cycle' should be removed.*

EC: Abstract: p1, L.25: is "global" necessary?

*AR: 'Total' has been removed on pg. 1, line 25 as well as in the caption of Figure 3, as 'global' is consistently used in the paper (pg. 1, line 25).*

EC: Abstract: p2, L.4-5: "At all other times... time varying" is this part of the abstract necessary? Please cut sentence in half after "time varying".

*AR: The first part of the sentence has been cut, and the sentence has been shortened (pg. 2, lines 5-8).*

EC: P2, l.6: shouldn't it be "played"? And maybe: "Since the Devonian, other feedback mechanisms....:"

*AR: We have made appropriate changes to the sentence at the Editor's recommendation (pg. 2, lines 5-7).*

EC: p3, L. 2: take out one "due to"

*AR: The repeated 'due to' was removed (pg. 3, line 2).*

EC: p8, L.7: "Ridgwell and Zeebe"

*AR: The author's name is now spelled correctly (pg. 8, line 7).*

EC: p9, L. 8: big error statement in bold.

*AR: We believe the Editor refers to the statement on pg. 10, lines 13-15:*

> *"Due to the complexity and time-variability of subduction, we do not consider times during which flat slab subduction may have occurred, which would result in greater arc-trench distances and is beyond the scope of this study."*

*Our approach of computing the statistical distribution of arc-trench distances at present (which, in turn, capture variations in slab dip), should also capture the variability of trench-arc distances through time. While our analysis will necessarily not capture 15% of arc distances that lie outside of our trench-arc inclusion distance, we believe we have made a statistically sound assumption.*

*We have re-written and expanded Section 2.4 (pg. 10, lines 1, 4-8, 11-15) because we believe we were statistically rigorous in determining the trench-arc distance that would capture a majority of carbonate intersections, and wish to acknowledge the logic and limitations of our analysis for transparency.*

EC: P9, L. 19: take out "s" at the end of "CO2 contentration"
*AR: This has now been corrected.*

EC: p11, L. 9, 13 and 14: I doubt the reference to MATLAB is appropriate here. You could simply state the method (Wavelet coherence, low pass filter...) and acknowledge the MATLAB toolboxes in the
"Code availability" section (I see this is already done).
*AR: We have removed references to MATLAB in the body of the manuscript, and have left them in the 'Code and data availability' section (pg.11, lines 9, 13-14).*

EC: P15, Legend of Figure 3 still mentions CIC and non CIC.
*AR: This has now been corrected.*

EC: P 15, L. 15: "Similar to the XWT...."
*AR: This has now been corrected (pg. 15, line 15).*

EC: p17, L. 11-13: something wrong with the grammatical structure of this sentence...
*AR: The sentence has been re-structured for clarity (pg. 17, lines 11-17).*

EC: p18, L. 8-9: something wrong with the grammatical structure of this sentence...
*AR: The sentence has been re-structured for clarity (pg. 18, lines 9-12).*

[revised manuscript text omitted]